# FoxO transcription factors actuate the formative pluripotency specific gene expression programme

Laura Santini [1,2], Saskia Kowald[1], Luis Miguel Cerron-Alvan[1,2], Michelle Huth [1,2], Anna Philina Fabing[1], Giovanni Sestini [2,3], Nicolas Rivron [3] & Martin Leeb [1] ✉

Naïve pluripotency is sustained by a self-reinforcing gene regulatory network (GRN) comprising core and naïve pluripotency-specific transcription factors (TFs). Upon exiting naïve pluripotency, embryonic stem cells (ESCs) transition through a formative post-implantation-like pluripotent state, where they acquire competence for lineage choice. However, the mechanisms underlying disengagement from the naïve GRN and initiation of the formative GRN are unclear. Here, we demonstrate that phosphorylated AKT acts as a gatekeeper that prevents nuclear localisation of FoxO TFs in naïve ESCs. PTEN-mediated reduction of AKT activity upon exit from naïve pluripotency allows nuclear entry of FoxO TFs, enforcing a cell fate transition by binding and activating formative pluripotency-specific enhancers. Indeed, FoxO TFs are necessary and sufficient for the activation of the formative pluripotency-specific GRN. Our work uncovers a pivotal role for FoxO TFs in establishing formative post-implantation pluripotency, a critical early embryonic cell fate transition.

Pluripotent cells can give rise to all specialised cells that form an adult organism. During mouse embryonic development, a population of naïve pluripotent cells arises in the pre-implantation epiblast around embryonic day E3.5–E4.5[1]. During the transition from pre- to post-implantation development, epiblast cells navigate through a continuum of pluripotency states. Starting from a naïve state with unrestricted potential, cells transit through a formative state to acquire competence for somatic and germ cell lineage specification, ultimately entering a primed state where they initiate expression of lineage markers[2–5].

The naïve pluripotent state of pre-implantation epiblast cells can be captured in vitro using mouse embryonic stem cells (mESCs)[6,7]. Maintenance of a homogeneous ground state of pluripotency requires the addition of two small molecule inhibitors to the culture media: PD0325901 (MEK1/2 inhibitor) and CHIRON (GSK3α/β inhibitor), collectively referred to as 2i[8]. mESCs cultured in 2i resemble the E4.5 pre-implantation epiblast in terms of epigenetic and transcriptional status[1,9–11]. Naïve identity is defined by the expression of a self-reinforcing gene regulatory network (GRN) that comprises the core pluripotency transcription factors (TFs) *Oct4* (gene name: *Pou5f1*) and *Sox2*, and naïve-specific TFs including *Nanog, Esrrb, Tbx3, Klf4, Klf5* and others[12–15].

A balanced interplay of several signalling inputs is responsible for the maintenance of the naïve-specific GRN[16]. The cytokine LIF (Leukaemia Inhibitory Factor) plays a key role in maintaining pluripotency and was the first identified exogenous factor that can support mouse ESC culture along with serum supplementation[17,18]. LIF mainly activates the JAK/STAT3 and PI3K/AKT pathways which are crucial to sustain naïve pluripotency[15]. Although the role of the JAK/STAT pathway has been intensively studied, the function of PI3K/AKT signalling in pluripotency has received much less attention[19]. Overexpression of a constitutively active form of AKT is sufficient to maintain mESCs in an undifferentiated state, even in the absence of LIF[20]. PI3K/AKT signalling is thought to support naïve pluripotency through inhibition of both the MEK/ERK and the GSK3 pathways[21–23], although the underlying

[1]Max Perutz Laboratories Vienna, University of Vienna, Vienna BioCenter, 1030 Vienna, Austria. [2]Vienna BioCenter PhD Program, Doctoral School of the University of Vienna, Medical University of Vienna, 1030 Vienna, Austria. [3]Institute of Molecular Biotechnology of the Austrian Academy of Sciences (IMBA), Vienna BioCenter, 1030 Vienna, Austria. ✉e-mail: martin.leeb@univie.ac.at

mechanisms are unclear. Furthermore, AKT signalling feeds directly into the naïve GRN by activating the expression of *Tbx3* and *Nanog*[15,24].

Exit from the naïve pluripotent state and initiation of formative pluripotency can be recapitulated in vitro by releasing cells from 2i inhibition into basal N2B27 medium. This change in conditions leads to loss of self-renewal in the naïve state and an irreversible commitment to differentiate approximately 48 h after 2i withdrawal[22]. The exit from naïve pluripotency results in the dismantling of the naïve-specific GRN and the establishment of a new formative state-specific GRN. This is accompanied by a profound shift in the signalling landscape: LIF and AKT signalling are reduced, concomitant with an increase in FGF/ERK activity[15,25,26]. We and others found that *Pten*, a negative regulator of AKT, is among the top hits in genetic screens for drivers of ESC differentiation, highlighting the importance of downregulating the PI3K/AKT pathway to ensure timely exit from the naïve state[27–31]. However, how exactly *Pten* regulates the exit from naïve pluripotency remains elusive.

In this study, we find that FoxO transcription factors are regulated by AKT and play a previously unrecognised but critical role in the transition from naïve to formative pluripotency. Our findings indicate that AKT acts as a gatekeeper by maintaining FoxO TFs in the cytoplasm in the naïve state. At the initiation of differentiation, elevated PTEN levels lead to a reduction in AKT signalling, allowing FoxO TFs to localise to the nucleus where they play a pivotal role in facilitating the transition from naïve to formative pluripotency by regulating a switch in operative GRNs. Our findings uncover an intricate mechanism that regulates the orderly transition between gene regulatory networks that determine distinct pluripotent states.

## Results

### PTEN controls pAKT for timely exit from naïve pluripotency

We previously found that mESCs lacking *Pten* exhibit a pronounced defect in the exit from naïve pluripotency[28]. Indeed, 24 h after 2i-removal in N2B27 medium (N24), *Pten* KO mESCs displayed higher Rex1-GFPd2 (Rex1-GFP) reporter activity than wild-type (WT) cells (Fig. 1a, b). *Rex1* is specifically expressed in the naïve state, and its downregulation coincides with irreversible commitment to differentiation[22,27]. This defect in exit from the naïve state in *Pten* KO ESCs was rescued by expressing *Pten* through transfection of a plasmid encoding 3xFLAG-PTEN (Supplementary Fig. 1a, b).

*Pten* mRNA and protein levels increase during exit from naïve pluripotency, while phospho-AKT (pAKT) levels are concomitantly reduced (Fig. 1c–e), suggesting that PTEN may promote the transition to formative pluripotency by decreasing AKT activity. In *Pten* KOs, phospho-AKT levels were significantly higher than in WT cells (Fig. 1d). Thus, we set out to delineate the molecular mechanism by which PTEN-mediated AKT inhibition might drive exit from naïve pluripotency.

When active, AKT phosphorylates several targets that play crucial roles in distinct cellular processes. Among AKT targets, TSC2 (Tuberous Sclerosis 2), GSK3 (Glycogen Synthase Kinase 3), and the FOXO (Forkhead box O) class of transcription factors are strong candidates for mediating changes in mESC differentiation states. Indeed, these factors have been identified as hits in genetic screens for differentiation drivers[27–31], suggesting that the AKT/mTORC1, AKT/GSK3 and AKT/FOXO signalling axes could all participate in the regulation of the transition from naïve to formative pluripotency.

Therefore, we started by investigating the involvement of AKT/mTORC1. AKT-mediated phosphorylation leads to TSC2 inhibition and consequent mTORC1 activation. mTORC1 is one of two distinct complexes containing the serine/threonine protein kinase mTOR and regulates essential cellular processes, including cell growth, protein synthesis and autophagy via phosphorylation of S6K, 4EBP-1 and ULK1, respectively[32]. ESCs lacking *Tsc2* retained higher Rex1-GFP levels at N24 compared to WT cells, similar to *Pten* KO cells[28] (Supplementary Fig. 1c, d).

To evaluate whether *Pten* acts through the AKT/mTORC1 axis to regulate the exit from naïve pluripotency, we inspected RNA sequencing (RNA-seq) data from *Pten* and *Tsc2* KO mESCs[28] (Supplementary Fig. 1e). Both KOs showed delayed downregulation of naïve and delayed upregulation of formative marker genes at N24, with more pronounced effects observed in *Tsc2* KOs (Fig. 1f). Both in 2i and at N24, *Pten* and *Tsc2* KOs deregulated a similar set of genes (differentially expressed genes, DEGs; Fig. 1g). In 2i, DEGs were enriched for terms associated with lysosomal and metabolic regulation, in line with the known role of mTORC1 in regulating those processes[27,31,32] (Supplementary Fig. 1f). Consistent with the observed naïve exit defect, genes upregulated at N24 were enriched for terms related to pluripotency (Supplementary Fig. 1g).

Next, we inspected the phosphorylation level of direct targets of mTORC1 in WT, *Pten* KO and *Tsc2* KO ESCs. Phospho-4EBP1 (p4EBP1) and phospho-S6K (pS6K) were similarly increased in both KOs, confirming that *Pten* or *Tsc2* deletion increases mTORC1 pathway activity. Addition of the mTORC1 inhibitor Rapamycin reduced mTORC1 activity in both KOs (Supplementary Fig. 1h, i).

We then assessed whether this reduction restored normal differentiation potential. Rapamycin treatment promoted faster downregulation of Rex1-GFP in WT ESCs, and resulted in a complete rescue of the differentiation defect of *Tsc2* KO cells (Fig. 1h), in line with previously published data[27]. In contrast, in *Pten* KO cells Rapamycin achieved only a partial rescue. The restoration of differentiation potential was specific for ESCs depleted for members of the PI3K/AKT/mTOR pathway. Naïve exit-defective ESCs lacking the WNT/GSK3 pathway effector *Tcf7l1*[22,28] did not restore Rex1-GFP downregulation kinetics upon Rapamycin treatment.

To evaluate the extent of phenotypic rescue of *Pten* KO cells by Rapamycin, we performed RNA-seq using WT, *Pten* KO and *Tsc2* KO mESCs, differentiated in presence of DMSO or Rapamycin (Supplementary Fig. 1j). Changes in naïve pluripotency expression upon Rapamycin treatment were TF-specific: While addition of Rapamycin restored *Nanog* expression to WT levels in both *Pten* and *Tsc2* KOs, *Esrrb* and *Klf5* expression remained elevated in *Pten* KOs compared to *Tsc2* KOs (Fig. 1i). Further corroborating an incomplete rescue of *Pten* KOs by mTORC1 inhibition, a set of 377 previously identified naïve pluripotency-specific genes[33] showed a significantly stronger reduction in expression levels upon Rapamycin treatment in *Tsc2* compared to *Pten* KOs (Fig. 1j, k). Together, these results highlight that the *Pten* KO phenotype is not exclusively determined by hyperactivity of mTORC1.

This prompted us to next investigate the role of AKT-mediated phosphorylation of GSK3 on Serine 9, which tags it for degradation and thereby stabilises β-catenin[22]. GSK3 phosphorylation was recently proposed to be crucial for maintaining pluripotency in *Pten* KO mESCs[23]. Consistently, phospho-GSK3 (pGSK3) levels are increased in *Pten* KO cells (Supplementary Fig. 1k). We hypothesised that if indeed deactivation of the GSK3-TCF7L1 axis of the WNT pathway leads to the differentiation defect in *Pten* KO cells, then the resulting WNT pathway hyperactivity should be epistatic to the pharmacological inhibition of GSK3 by CHIRON. Such an epistatic interaction was observed in *Tcf7l1* KOs, where the activity of the β-catenin destruction complex is rendered obsolete and, hence, the addition of CHIRON had no additional effect (Supplementary Fig. 1l). In contrast, treatment of *Pten* KOs with CHIRON resulted in delayed differentiation speeds akin to WT cells. Furthermore, we did not previously observe a transcriptional signature typical of increased WNT activity in *Pten* KO ESCs[28]. This suggests that AKT-hyperactivity-dependent phosphorylation of GSK3 in *Pten* mutants has little causative impact on the exit from naïve pluripotency in our cellular system.

Collectively, these data show that *Pten* regulates pathways in addition to mTORC1 that are relevant for proper exit from naïve pluripotency. As the role of GSK3 appeared minor, we

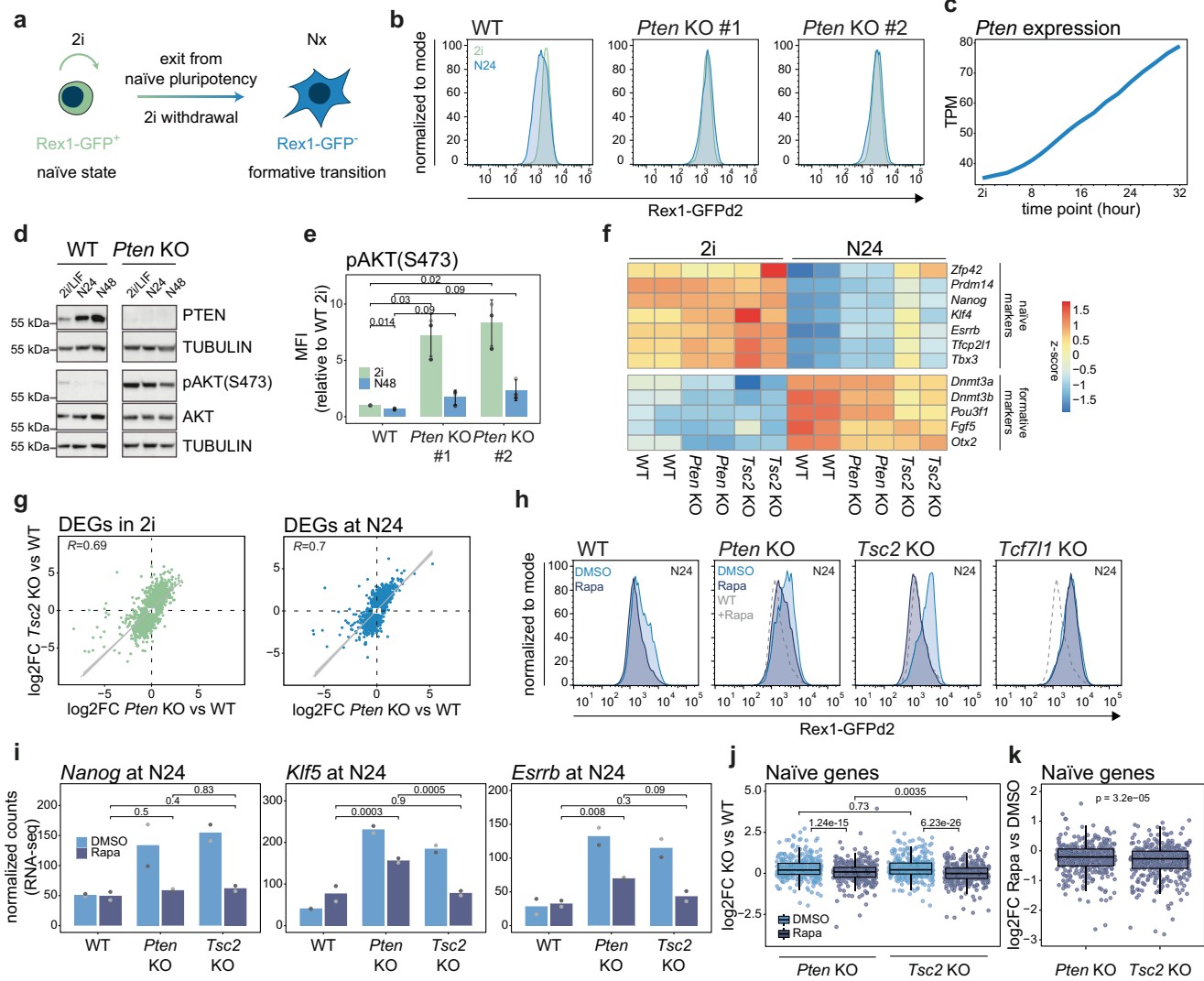

**Fig. 1 | mTORC1 inhibition only partially rescues *Pten* KO phenotype.**
**a** Schematic representation of our experimental setup. Nx indicates differentiation time (x) after 2i withdrawal in N2B27 medium. **b** Flow cytometry analysis of Rex1-GFP levels in WT and two independent *Pten* KO clones in naïve pluripotency supporting conditions (2i, green) and 24 h after 2i removal (N24, blue). One representative of $n = 11$ independent experiments is shown. **c** *Pten* expression levels (transcript per million, TPM) from a 2 h-resolved RNA-seq differentiation time course in WT[28]. **d** Western blot analysis for indicated proteins in WT and *Pten* KO in 2i/LIF, at N24 and N48. TUBULIN served as a loading control. **e** Quantification of pAKT (S473) levels by flow cytometry in indicated cells and conditions. Mean and standard deviation (SD) of mean fluorescence intensity (MFI) from $n = 3$ independent experiments (distinct shades of grey) are shown. *p* values result from paired, two-tailed *t*-tests. **f** Heatmap showing row-normalised expression values of indicated naïve and formative genes in indicated genotypes and conditions. **g** Scatter plots showing correlation between differentially expressed genes (DEGs, *p* adj. ≤

0.05, |log2foldchange (log2FC)| ≤ 0.5) in *Pten* or *Tsc2* KO in 2i and N24. Trend lines with 95% confidence intervals, and Pearson's correlation coefficients (*R*) are shown. **h** Flow cytometry analysis of Rex1-GFP levels in indicated cell lines at N24, after indicated treatments. Rapamycin-treated WT cells are shown as dashed grey line. One representative of $n = 5$ independent experiments is shown. **i** Expression levels of indicated genes in indicated cell lines at N24 after indicated treatments measured by RNA-seq. *p* adj. values resulting from DESeq analysis are indicated in the plot. **j** Box plot showing expression of 377 naïve genes (naïve early and naïve late genes[33]) in indicated KOs and conditions measured by RNA-seq. *p* values from two-tailed Wilcoxon signed rank tests are indicated. **k** Box plot showing the expression of 377 naïve genes in indicated KOs and conditions measured by RNA-seq. *p* value from two-tailed Wilcoxon signed rank test is indicated. All box plots in this paper show the 25th and 75th percentiles and median. Whiskers indicate minimum and maximum values.

focussed our attention on FoxO TFs, the third signalling axis downstream of AKT.

**FoxO nuclear translocation promotes naïve pluripotency exit**
FoxO TFs regulate several crucial cellular processes, including cell cycle, apoptosis and DNA repair[34]. AKT-mediated phosphorylation retains FoxO TFs in the cytoplasm. In both WT and *Pten* KO ESCs cultured in 2i, immunofluorescence (IF) analysis revealed a largely cytoplasmic localisation of FOXO1. Upon exit from the naïve state in WT cells, we observed that FOXO1 translocated to the nucleus, concomitant with reduced ESRRB levels (Fig. 2a, b) and a

downregulation of AKT activity (Fig. 1d, e)[35]. Increased nuclear FOXO1 levels could be observed as early as 8 h after 2i withdrawal (N8) and increased further until N24. This nuclear translocation of FOXO1 at the onset of ESC differentiation was severely impaired in *Pten* KO cells, which maintained a clear cytoplasmic FOXO1 localisation at N24 (Fig. 2a, c). Nucleo-cytoplasmic fractionation experiments showed similar results for FOXO3 (Supplementary Fig. 2a). Sequestration of FOXO1 in the cytoplasm is regulated through pAKT-mediated FOXO1 phosphorylation (pFOXO1). Consistently, we observed a decrease in pFOXO1 concomitant with an increase in total FOXO1 during the exit from naïve pluripotency in WT. In *Pten*

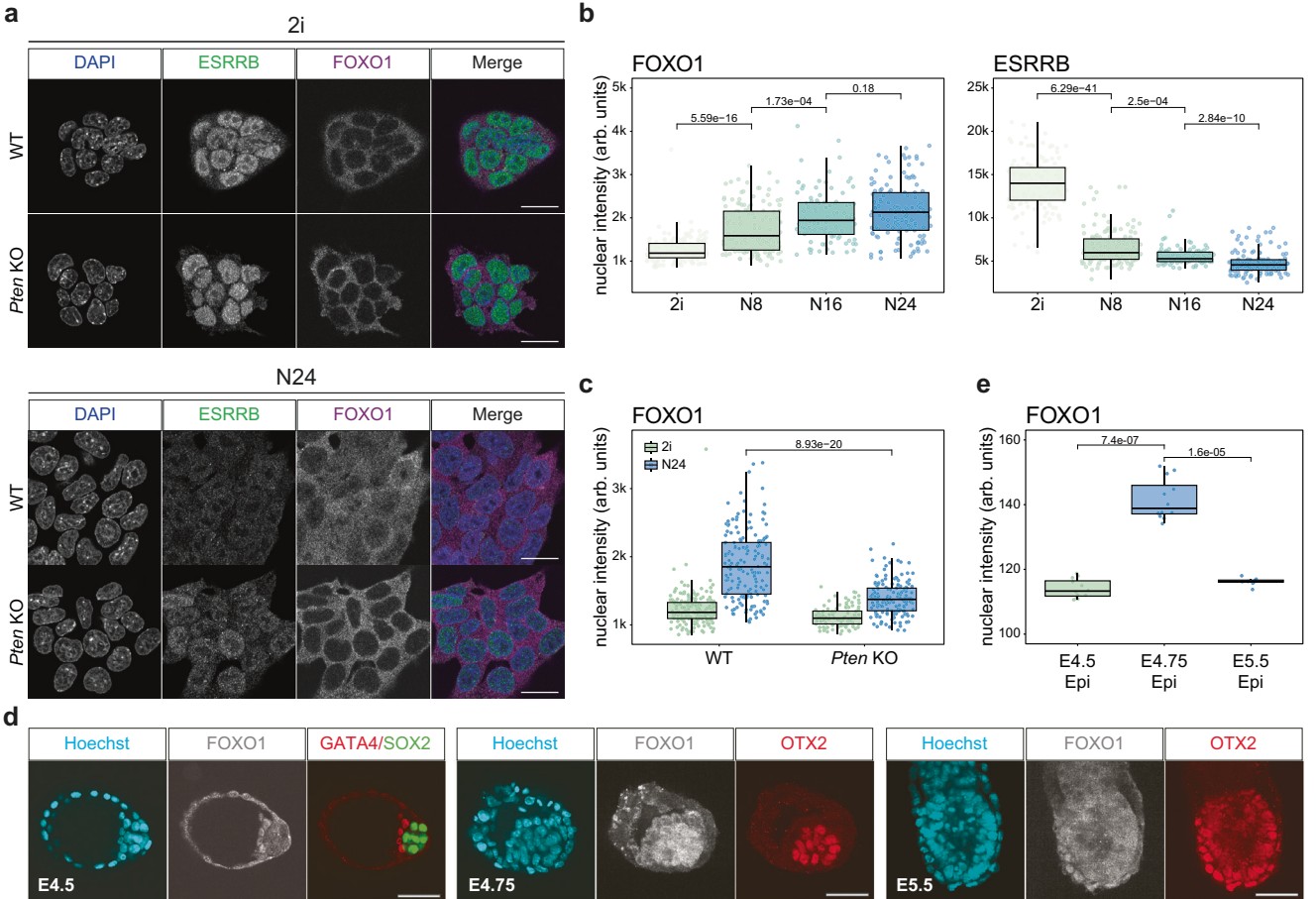

**Fig. 2 | FoxO TFs translocate into the nucleus upon exit from naïve pluripotency.**
**a** Confocal microscopy images after IF, showing FOXO1 (purple) and ESRRB (green) in WT and *Pten* KOs in 2i and at N24. DAPI staining is shown in blue. One representative image from *n* = 3 independent experiments is shown. Scale bar = 20 µM. **b** Quantification of FOXO1 and ESRRB nuclear intensity (in arbitrary unit, arb. units) measured from confocal images of WT cells in 2i (*n* = 135) and 8 h (N8, *n* = 136), 16 h (N16, *n* = 88) and 24 h (N24, *n* = 127) after 2i withdrawal. Data were obtained from *n* = 2 independent experiments. *p* values are derived from a two-tailed Wilcoxon rank sum test. **c** Quantification of FOXO1 nuclear intensity (arb. units) based on images of WT (*n* = 189 [2i] and *n* = 164 [N24]) and *Pten* KO (*n* = 100 [2i] and *n* = 153

[N24]), as in (**a**). Data were obtained from *n* = 2 independent experiments. *p* values were calculated as in (**b**). **d** Confocal microscopy images after IF showing FOXO1 (white), SOX2 (green), GATA4 (red), or OTX2 (red) in WT E4.5, E4.75 and E5.5 embryos. Hoechst staining is shown in cyan. One representative image of *n* = 3 independent experiments is shown. Scale bar = 100 µM. **e** Quantification of FOXO1 nuclear intensity (arb. units) measured from confocal images of WT epiblast from E4.5 (*n* = 12), E4.75 (*n* = 12) and E5.5 (*n* = 8) embryos, as in (**d**). Data were obtained from *n* = 3 independent experiments. Indicated *p* values are derived from a two-tailed Wilcoxon rank sum test.

mutant cells pFOXO1 levels are elevated in 2i and at N24 (Supplementary Fig. 2b).

Supporting a functional relevance of nuclear shuttling of FoxO TFs in vivo, we detected transient nuclear FOXO1 in the OTX2-positive epiblast of E4.75 embryos. This was not detectable in E4.5 or in E5.5 epiblast cells (Fig. 2d, e and Supplementary Fig. 2c). This is consistent with the hypothesis that FOXO1 nuclear shuttling is involved in the initiation of formative pluripotency in in vivo peri-implantation development.

To test whether nuclear translocation of FOXO1 was indeed AKT-dependent, we analysed the nuclear vs. cytoplasmic localisation of FOXO1 after treatment with the specific allosteric AKT inhibitor MK-2206[36]. MK-2206 treatment elevated levels of nuclear FOXO1, in both 2i and at N24, and resulted in expedited downregulation of Rex1-GFP in WT cells (Supplementary Fig. 2d–f), consistent with a requirement for reducing AKT activity to enable and possibly trigger the exit from naïve pluripotency.

Our results support the hypothesis that nuclear translocation of FoxO1 is crucial to induce proper exit from naïve pluripotency. To test this hypothesis, we performed doxycycline-induced expression of a

constitutively nuclear version of FoxO1 (3xFLAG-FoxO1[nuc])[37]. This treatment was sufficient to extinguish naïve pluripotency in WT cells cultured in 2i within 8 h in a dose-dependent manner (Supplementary Fig. 2g–j). Underscoring a causal link between aberrant localisation of FOXO1 in *Pten* mutant ESCs and defects in naïve exit, 3xFLAG-FoxO1[nuc] expression in *Pten* KO ESCs rescued their differentiation defect (Supplementary Fig. 2k). Interestingly, overexpression of nuclear FoxO1 in *Pten* KO cells had distinct results on different naïve markers: While it reduced the expression of *Nanog* and *Esrrb*, in line with the global rescue of the differentiation defect of *Pten* mutants, it upregulated the expression of *Klf5* in a dose-dependent manner (Supplementary Fig. 2l). This indicates that FoxO1 can exert target-specific activator or silencer function. We also noted that prolonged exposure to high levels of nuclear FoxO1 (>24 h) was cytotoxic, probably due to induction of the pro-apoptotic programme by FoxO1[38].

In summary, our data show that AKT-mediated nuclear translocation of FoxO1, and possibly other FoxO TFs, is essential for the timely exit from naïve pluripotency. We conclude that lack of nuclear translocation of FoxO TFs underlies, at least in part, the differentiation defect observed in *Pten* KO mESCs.

## FoxO TFs bind enhancers activated in formative pluripotency

To explore the molecular role of FoxO TFs in the transition from naïve to formative pluripotency, we performed chromatin immunoprecipitation followed by next-generation sequencing (ChIP-seq) analysis for FOXO1 and FOXO3 in WT and *Pten* KO cells. This analysis was performed in 2i and at N24, that is, 24 h into the formative transition (Fig. 3a and Supplementary Fig. 3a–c). In WT cells, we observed a strong increase in the number of genomic loci bound by FOXO1 and FOXO3 at N24 compared to 2i. We detected 1314 and 391 peaks in 2i and 2840 and 623 peaks at N24 for FOXO1 and FOXO3, respectively. This is consistent with increased nuclear localisation of FoxO TFs at the exit from naïve pluripotency. In agreement with the largely cytoplasmic localisation in the absence of *Pten*, FOXO TF ChIP-seq signals were barely detectable in *Pten* KO cells (Supplementary Fig. 3b, c). Of note, due to low amounts of precipitated DNA in 2i, ChIP-seq library-prep will most likely have amplified signal in 2i relative to N24. Hence, a direct quantitative comparison between 2i and N24 FOXO TF signal is not possible.

FOXO1 and FOXO3 ChIP-seq data showed a significant overlap. Almost 50% of FOXO3 peaks overlapped with FOXO1 peaks, and FOXO1 signal was detected at virtually all FOXO3 peaks (Supplementary Fig. 3c, d). Based on this finding, we performed all downstream analysis with the group of FOXO1-bound loci, if not otherwise stated. We then divided FOXO1-bound regions into three groups depending on peak-calling results: 2i-only peaks (612), N24-only peaks (2138) and shared peaks (702) (Fig. 3a). KEGG and REACTOME pathway enrichment analyses (EA) showed an enrichment for ubiquitin mediated proteolysis in the 2i-specific peaks and for PI3K/AKT signalling, focal adhesion and actin cytoskeleton related terms in the N24 specific peaks. Shared peaks were enriched for terms related to general pluripotency and TGF-β signalling (Supplementary Fig. 3e and Supplementary Data 1). All peak categories were enriched for FoxO-motifs (Supplementary Data 1).

FoxO TF peaks were located mainly outside of promoter regions (Supplementary Fig. 3f), indicating a potential contribution of FoxO TFs to enhancer regulation. To test this, we utilised published ChIP-seq datasets[39] for the enhancer marks H3K27ac and p300, obtained in ESCs and in epiblast-like cells (EpiLCs). EpiLCs represent the in vitro counterpart of formative epiblast cells of the E5.5 blastocyst[40] and correspond to a developmental state akin to mESCs at N24. We found that strong H3K27ac and p300 signals in EpiLCs overlapped with the regions we identified as bound by FoxO TFs at N24 (Fig. 3b). Moreover, regions bound by FoxO TFs in 2i were significantly enriched in ESC-specific enhancers and enhancers active in ESCs and EpiLCs[41] (Fig. 3c), whereas EpiLC-specific enhancers were not significantly bound by FOXO1 in 2i. In contrast, N24 FOXO1-peaks significantly overlapped with shared and EpiLC-specific enhancers while showing no significant overlap with ESC-specific enhancers. A striking total of 74% of FOXO1 peaks at N24 are found on EpiLC enhancers. Consistently, we detected a significant enrichment of FoxO1 and FoxO3 motifs in EpiLC-specific enhancers (Supplementary Fig. 3g). This is consistent with a role for FoxO TFs in activating formative-specific regulatory regions upon exit from the naïve state.

To assess a potential role of AKT signalling in mediating chromatin accessibility, we performed assay for transposase-accessible chromatin using sequencing (ATAC-seq) in WT and *Pten* KO cells, in 2i and at N24 (Fig. 3d and Supplementary Fig. 3h). Regions that showed FoxO TF binding exclusively in 2i were open only in the naïve state and showed weak ATAC-seq signal at N24. In contrast, regions exhibiting FoxO TF binding at N24 showed near equal ATAC-seq signal in both 2i and at N24. At the resolution of our analysis, ATAC-seq signal was indistinguishable between WT and *Pten* KO cells, in which FoxO TF translocation to the nucleus is largely abolished. This suggests that FoxO TFs are unlikely to act as pioneer factors that open chromatin during the naïve to formative transition. Instead, our data is consistent with a role of FoxO TFs in activating already open poised chromatin as suggested before[42] also during the exit from naïve pluripotency.

OCT4 and OTX2 are key regulators of general and formative pluripotency, respectively. OTX2 causes relocation of OCT4 from ESC- to EpiLC-specific enhancers upon the exit from the naïve state[39]. FOXO1 2i peaks showed a significant but relatively weak overlap with OCT4-bound ESC enhancers, and 4% of all OCT4-bound ESC enhancers were FoxO1 targets (Fig. 3e, f). In contrast, OCT4-bound EpiLC enhancers showed a stronger and highly significant overlap with FOXO1 at N24, and 12% of OCT4-bound EpiLC-enhancers were FoxO1 targets at N24 (Fig. 3f). A similarly large portion of 14% of OTX2-bound EpiLC enhancers were bound by FOXO1 at N24 (Fig. 3g, h). Moreover, our analysis clearly showed that colocalization of FOXO1 with OCT4 and OTX2 occurs nearly exclusively on enhancers, suggesting that FoxO TFs regulate enhancer activation in cooperation with pluripotency-state specific TFs.

Recently it was shown that *Esrrb*, originally identified as a naïve pluripotency-specific TF, performs additional functions in the initiation of the formative transcription programme[33]. To investigate whether FoxO TFs cooperate with ESRRB, we examined the overlap on chromatin between FOXO1 and ESRRB throughout the pluripotency continuum (Fig. 3i, j). These analyses showed that a significant 35% of 2i-specific and 38% of shared FoxO1 peaks are also bound by ESRRB in 2i (Fig. 3j). Regions bound by ESRRB at N48 still showed a highly significant overlap with FOXO1 binding, with 28% of shared FoxO1 peaks also decorated by ESRRB at N48. These FOXO1-ESRRB co-bound loci were in close proximity to, and thus potentially regulate, known formative (*Otx2, Dnmt3a, Lef1*) and naïve marker genes (*Nanog, Esrrb, Zfp42, Klf2, Klf4, Klf5, Tfcp2l1, Nr5a2, Prdm14*) (Supplementary Data 2). Co-bound regions consistently show an overall stronger binding intensity of FOXO1 and ESRRB and also exhibit higher overall H3K27ac and p300 levels compared to loci bound by only FOXO1 or only ESRRB (Supplementary Fig. 3i). We also noted that regions only bound by ESRRB exhibited weaker ATAC-seq signal, indicating that ESRRB requires additional factors such as FOXO1 to achieve full chromatin activation. FOXO1-only bound regions, in turn, showed clear enhancer activity in 2i and at N24, suggesting that FOXO1 can work independently of ESRRB.

FoxO TFs have been shown to interact with the WNT pathway effector β-catenin to enhance transcriptional output[43]. In line with this, a highly significant 50% of all FOXO1-bound regions were also occupied by β-CATENIN (Fig. 3k), suggesting a functional interaction of FoxO TFs and β-catenin during the transition to formative pluripotency.

In sum, our findings reveal that FoxO TFs bind to a large set of enhancers that are activated upon the exit from naïve pluripotency. FOXO1 cooperates with core components of the naïve and formative TF-repertoire. We hypothesise that this is to ensure faithful firing of the formative GRN.

## FoxO TFs instruct rewiring of the naïve to the formative GRN

We next sought to identify the transcriptional consequences of FoxO TF induced changes to chromatin at the exit from naïve pluripotency. We specifically wanted to know which components of the formative state-specific GRN might be functionally dependent on regulation by FoxO TFs. To this end, we compared the changes in transcript levels upon exit from naïve pluripotency[28] between distinct sets of FoxO1-bound genes as defined above. Overall, FoxO1 targets were highly enriched in genes differentially expressed within the first 24 h of ESC differentiation (Supplementary Fig. 4a). 2i-specific FoxO1 targets were overall downregulated at N24, while genes bound by FOXO1 at N24 showed an overall upregulation during naïve exit (Fig. 4a and Supplementary Fig. 4b). The expression of targets bound in both conditions (2i&N24) did not show a strong and consistent directional change in gene expression during the exit from naïve pluripotency. To further investigate the expression kinetics of FoxO1 target genes

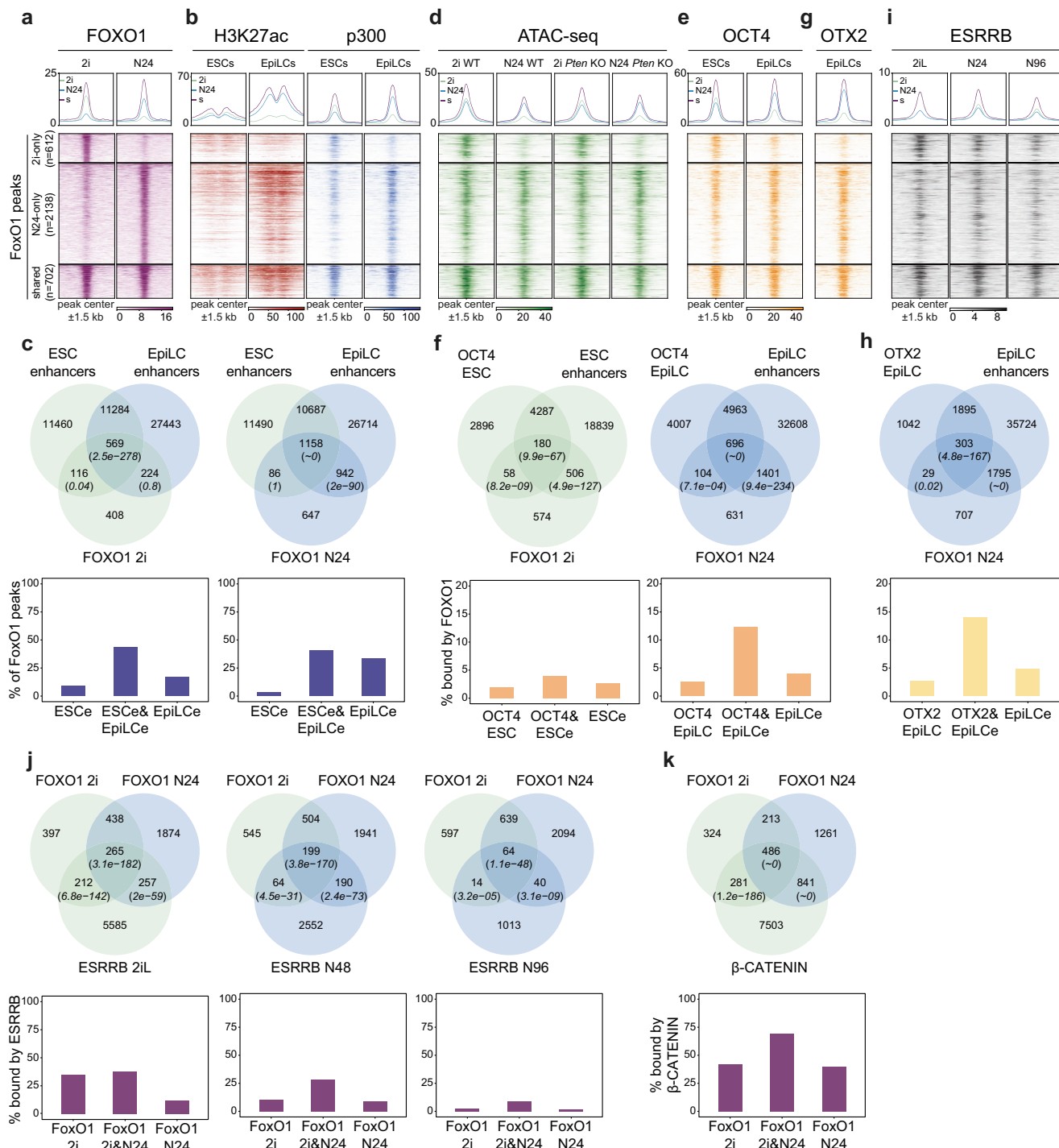

**Fig. 3 | Chromatin dynamics of FoxO TFs upon exit from the naïve pluripotent state. a** Heatmap displaying FoxO1 signal within a 1.5 kb window around FoxO1 peak-centres in WT cells at 2i and N24, categorised into 2i-only (n = 612, green), N24-only (n = 2138, blue) and shared peaks (n = 702, purple). **b** Heatmaps showing H3K27ac (left) and p300 (right) signal in ESCs and EpiLCs[39] in a 1.5 kb window around FoxO1 peak-centres. **c** Venn diagrams showing overlaps between FoxO1 peaks in 2i and N24 with ESC and EpiLC enhancers (top). Bar plots display the percentage of FoxO1 peaks overlapping with ESC-specific (ESCe), EpiLC-specific (EpiLCe) or shared (ESCe&EpiLCe) enhancers (bottom). p values are derived from hypergeometric tests. **d** Heatmap showing ATAC-seq signal within a 1.5 kb window around FoxO1 peak-centres in WT and Pten KOs in 2i or at N24. **e** Heatmaps showing Oct4 signal in ESCs and EpiLCs[39] in a 1.5 kb window around FoxO1 peak-centres. **f** Venn diagrams showing the overlap between FoxO1 peaks in 2i or at N24 with Oct4 peaks in ESCs or EpiLCs (top). Bar plots showing the

percentage of OCT4-bound enhancers overlapping with FoxO1 peaks (bottom). p values are derived from hypergeometric tests. **g** Heatmap showing Otx2 signal in EpiLCs[39] in a 1.5 kb window around FoxO1 peak-centres. **h** Venn diagram showing the overlap between FoxO1 peaks at N24 with Otx2 peaks in EpiLCs (top). Bar plots showing the percentage of OTX2-bound enhancers overlapping with FoxO1 peaks (bottom). p values are derived from hypergeometric tests. **i** Heatmap showing Esrrb signal in WT cells in 2iL, N48 and N96[33] in a 1.5 kb window around FoxO1 peak-centres. **j** Venn diagrams showing the overlap between FoxO1 peaks in 2i and at N24 with Esrrb peaks in 2iL, N48 or N96 (top). Bar plots showing the percentage of FoxO1 peaks (2i-only, N24-only or shared) that overlap with Esrrb peaks (bottom). p values are derived from hypergeometric tests. **k** Venn diagrams showing the overlap between FoxO1 peaks in 2i and at N24 with β-catenin peaks (top). Barplots showing the percentage of FoxO1 peaks in indicated categories overlapping with β-catenin peaks (bottom). p values are derived from hypergeometric tests.

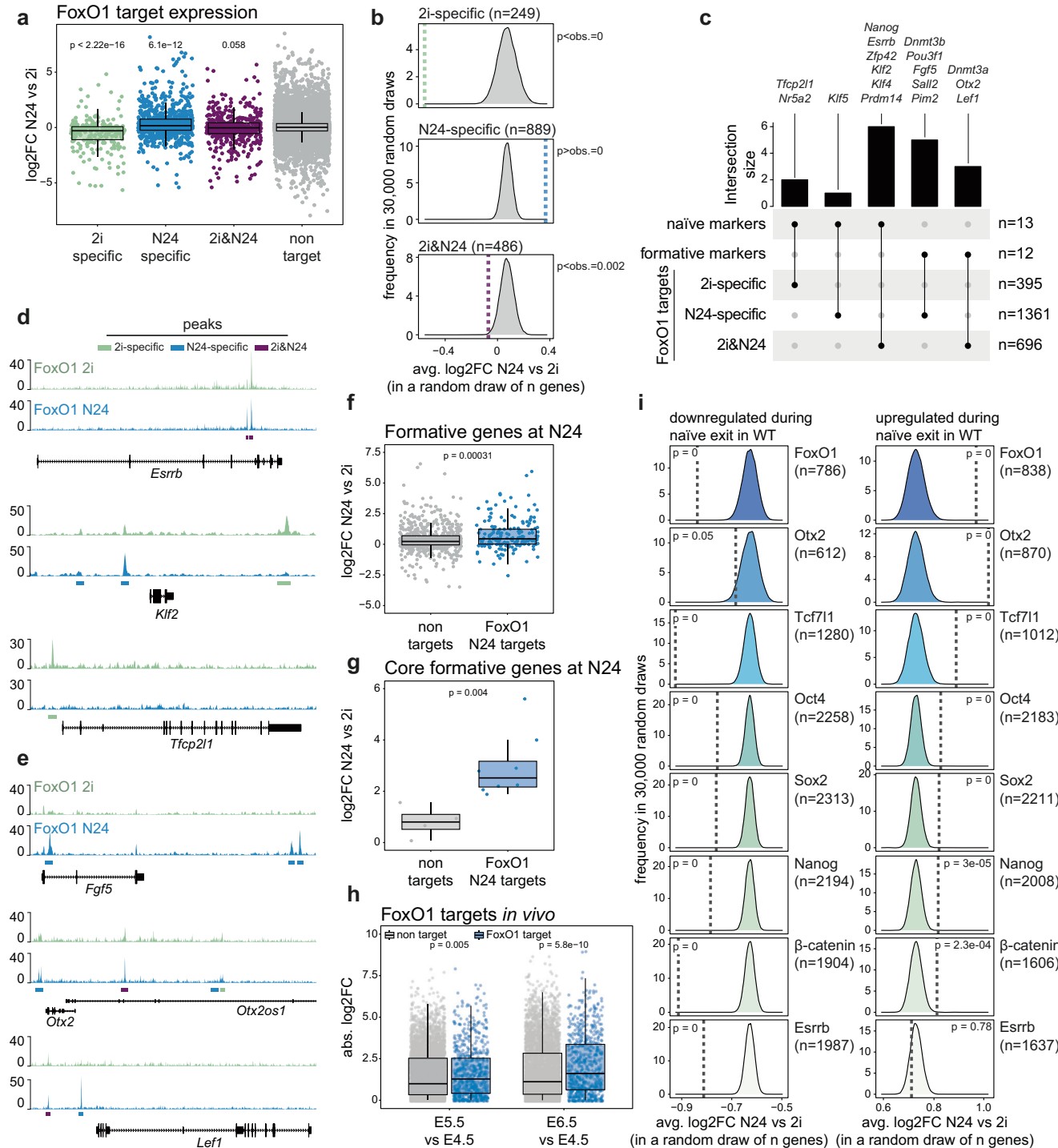

**Fig. 4 | FoxO TF targets are key players in the naïve to formative pluripotency transition. a** Boxplot illustrating expression of 2i-specific ($n = 249$, green), N24-specific ($n = 889$, blue), and 2i&N24 ($n = 486$, purple) FoxO1 targets in WT cells at N24, measured by RNA-seq. Data show log2FC relative to 2i and $p$ values from Wilcoxon rank sum tests. Non-targets are shown in grey. **b** Bootstrapping analysis (30,000 draws) for gene-sets of the same size as the sample gene-sets (2i-specific, 249 genes; N24-specific, 889 genes; 2i&N24, 486 genes). The distribution of average log2FC is plotted, with $p$ values indicating the probability of the predicted exceeding the measured effect-size. **c** Upset plot showing the overlap of FoxO1 targets with core naïve and formative markers. **d** Genome browser snapshots displaying FoxO1 ChIP-signal in WT on selected naïve markers in 2i (green) and N24 (blue). Called peaks are indicated (2i-only: green, N24-only: blue, shared: purple). **e** Genome browser snapshots showing FoxO1 ChIP-signal in WT cells on selected formative markers in 2i (green) and N24 (blue). Called peaks are indicated (2i-only:

green, N24-only: blue, shared: purple). **f** Box plot depicting expression of formative genes ($n = 832$) in WT cells at N24, divided into FoxO1 N24 targets (blue) or non-targets (grey), as measured by RNA-seq. Data are shown as log2FC relative to 2i. $p$ value from two-tailed Wilcoxon rank sum tests is indicated in the plot. **g** Box plot showing the expression of core formative genes ($n = 12$) in WT cells at N24, divided into FoxO1 N24 targets (blue) or non-targets (grey), measured by RNA-seq. Data are shown as log2FC relative to 2i. $p$ value from two-tailed Wilcoxon rank sum test is indicated. **h** Box plot showing expression of FoxO1 targets ($n = 726$, blue) and non-targets ($n = 5061$, grey) in E5.5 and E6.5 embryos[28,68]. Absolute log2FC relative to E4.5 are shown. $p$ values from two-tailed Wilcoxon rank sum test are indicated. **i** Similar to (**b**), bootstrapping analysis (30,000 draws) for gene-sets of the same size as the sample gene-sets ($n$ = group size). The distribution of average log2FC is plotted, with $p$ values indicating the probability of the predicted exceeding the measured effect-size.

during the exit from naïve pluripotency, we followed their transcript levels across a 32 h differentiation time course at a 2 h-resolution[28]. We found that 2i-specific FoxO1 targets showed a largely continuous downregulation, N24-specific targets a continuous upregulation, and 2i&N24 targets a lack of clear directionality (Supplementary Fig. 4c).

To further determine whether the observed gene expression changes were stronger than expected by chance, we computed the distribution of average log2 fold changes (log2FC) across 30,000 randomly sampled gene groups. Sampled gene groups were matched in size to different FoxO1 target groups (Fig. 4b) and individual genes were drawn from RNA-seq data comparing expression between 2i and N24. The results showed that 2i- and N24-specific FoxO1 targets by far exceed the levels of gene expression changes observed in any randomly sampled gene group. Specifically, FoxO1 2i targets are highly significantly more downregulated, and N24 FoxO1 targets are highly significantly more upregulated during the exit from naïve pluripotency than randomly selected gene sets. These findings suggest that FoxO1 binding is a major determinant of gene expression changes during differentiation.

Recent work reported transcriptome analysis of differentiating mESCs up to 96 h after 2i withdrawal and defined 6 distinct groups based on gene expression kinetics: naïve early and naïve late (downregulated early or late upon naïve exit), formative early and formative late (upregulated during naïve exit), committed early and committed late (upregulated late during naïve exit)[33]. We found that 2i-specific FoxO1 targets were enriched for naïve and formative early genes, whereas N24-specific and 2i&N24 FoxO1 targets were enriched for naïve early, formative (early and late) and committed early genes (Supplementary Fig. 4d).

As these results indicated a link between FoxO TFs and central components of the naïve and formative GRNs, we further tested whether FoxO1 binds and potentially regulates core members of the naïve or formative GRN. We found that 9 out of 13 core naïve marker genes (modified from ref. 44) were bound by FoxO1 in 2i (*Tfcp2l1*, *Nr5a2*), in both 2i and at N24 (*Nanog*, *Esrrb*, *Zfp42*, *Klf2*, *Klf4*, *Prdm14*) or only at N24 (*Klf5*) (Fig. 4c, d and Supplementary Data 2). N24-specific binding of *Klf5* might explain why this naïve marker gene is further up- instead of down-regulated, when nuclear FOXO1 is expressed in *Pten* KOs during differentiation (Supplementary Fig. 2l). Conversely, 8 out of 12 key formative marker genes[44] (hereafter referred to as core formative genes) were bound by FoxO1 at N24 (*Dnmt3b*, *Pou3f1*, *Fgf5*, *Sall2*, *Pim2*) or under both conditions (*Dnmt3a*, *Otx2*, *Lef1*) (Fig. 4c, e and Supplementary Fig. 4e). Notably, formative genes (early and late, as described above) and core formative genes that are direct FoxO1 targets showed a significantly stronger upregulation during the exit from naïve pluripotency than non FoxO1-targets (Fig. 4f, g).

Notably, in vivo FoxO1 targets are upregulated more strongly than non-targets between E4.5 and E5.5 (Fig. 4h). This is consistent with our previous results showing high nuclear FoxO1 levels in E4.75 blastocysts (Fig. 2d, e), and we hypothesise that transient nuclear localisation of FoxO1 in rosette-stage blastocysts at E4.75 facilitates the establishment of the formative post-implantation like pluripotency programme.

Among the FoxO1 targets, we identified a significant number of genes that were also found in a genetic screen for genes controlling the exit from naïve pluripotency, here referred to as 'exit factors' (Supplementary Fig. 4f)[28,29]. Among exit factors that are upregulated during naïve exit, FoxO1 targets showed significantly stronger regulation (Supplementary Fig. 4g). This suggests that FoxO TFs might function by facilitating the activity of multiple processes required for proper differentiation.

In total, 22% of genes differentially expressed between WT and *Pten* KOs at N24 were FoxO1 target genes (Supplementary Fig. 4h). Furthermore, those components of the core formative GRN that are FoxO1 targets exhibited a stronger deficiency in upregulation compared to non-targets in *Pten* mutant cells at N24 (Supplementary

Fig. 4i). Further highlighting a causal link between inactivation of the FoxO-signalling axis in *Pten* KOs and the molecular defects observed in these mutants, FoxO1 2i target genes showed a significantly stronger downregulation compared to non-targets in *Pten* KO ESCs (Supplementary Fig. 4j, k).

To finally assay the impact of FoxO1 relative to other known master regulators of pluripotency, we again performed bootstrapping experiments, akin to the ones described above (Fig. 4b). We generated the distribution of average log2FC during naïve exit across 30,000 random sampled groups of genes, each matched to the size of the respective TF-target group (Fig. 4i). Our analyses show that FoxO1 targets exhibit both a stronger positive and negative transcriptional response during naïve exit than targets of well-established pluripotency TFs, such as OCT4, SOX2, ESRRB or NANOG. Only OTX2-bound genes exhibited a slightly stronger upregulation and TCF7L1- and β-CATENIN bound genes a stronger downregulation than FoxO1 target genes. Notably, even within OTX2-bound genes, genes co-bound by OTX2 and FOXO1 showed the strongest transcriptional upregulation, suggesting an interaction between OTX2 and FOXO1 at the initiation of the formative GRN (Supplementary Fig. 4l). Genes co-bound by FOXO1 and TCF7L1 in ESCs showed a significantly stronger transcriptional downregulation during differentiation compared to single factor targets and non-targets (Supplementary Fig. 4m), suggesting a cooperative role of FOXO1 and TCF7L1 in silencing the naïve pluripotency specific GRN. Together, these data support a dual role for FoxO1 in activation of the formative and silencing of naïve specific genes, depending on the chromatin context.

In sum, our data suggest that FoxO1 acts as a key regulator of the naïve to formative transition by binding to and regulating major components of the naïve and formative-specific GRNs. This function is potentially performed in cooperation with core pluripotency TFs such as OCT4, ESRRB, NANOG, TCF7L1 and OTX2 which are known to regulate multiple pluripotency states and transitions. Our analysis further indicates that the effect size of FoxO1 regulation surpasses that of most other core pluripotency TFs. Together, this positions FoxO1 as central TF to regulate the transition from naïve to formative pluripotency.

**FoxO TFs are required for proper exit from naïve pluripotency**

To investigate a potential requirement for *FoxO1* for the transition from naïve to formative pluripotency, we analysed the differentiation capacity of ESCs after FoxO1 depletion. To this end, we turned to knockdown (KD) experiments using short-interfering RNAs (siRNAs). Treatment with siRNAs against *FoxO1* resulted in the downregulation of transcript and protein levels (Supplementary Fig. 5a, b). Cells with reduced levels of *FoxO1* (siFoxO1) retained higher expression of Rex1-GFP at N24 compared to cells treated with scrambled siRNAs (siScr), suggesting that *FoxO1* is required for proper exit from naïve pluripotency (Fig. 5a).

KEGG pathway enrichment analysis after RNA-seq analysis of FoxO1-siRNA treated cells at N24 showed that upregulated genes were enriched for signalling pathways associated with pluripotency maintenance (Supplementary Data 1), confirming the differentiation defect. Further consistent with the differentiation delay upon FoxO1 depletion, we observed that naïve pluripotency markers failed to be down- and formative markers upregulated upon FoxO1 KD (Supplementary Fig. 5c). Both genes that were up- and downregulated upon FoxO1 depletion were highly enriched in FoxO1 ChIP-seq targets (Fig. 5b and Supplementary Fig. 5d), with 30% and 25%, respectively, being bound by FOXO1.

Overall, genes up- or downregulated in *Pten* KOs at N24 were also up- or downregulated in siFoxO1-treated cells at N24 (Supplementary Fig. 5e) and FoxO1 target genes showed a stronger deregulation in the absence of either *Pten* or *FoxO1* compared to non-targets (Supplementary Fig. 5f). This further supports the proposition that the *Pten* KO phenotype is caused, in part, by misregulated FoxO TF localisation.

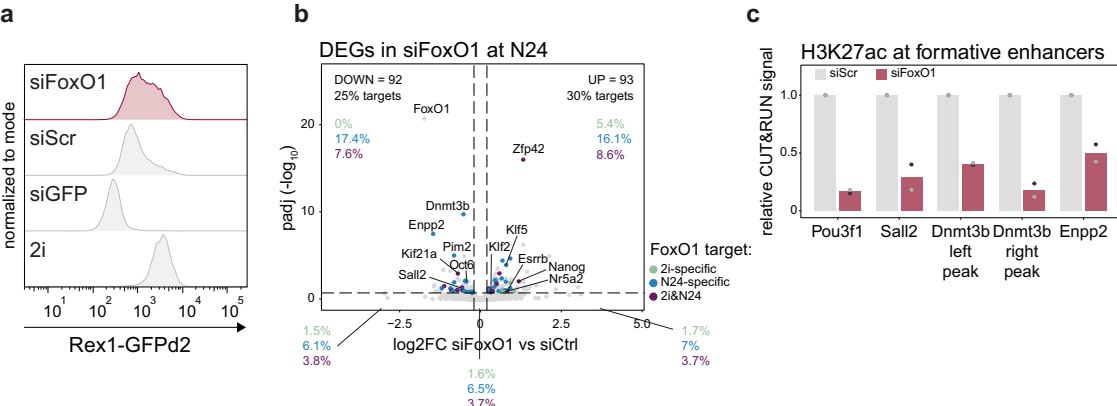

**Fig. 5 | Interference with FoxO1 nuclear shuttling impairs the transition from the naïve to the formative GRN. a** Flow cytometry analysis of Rex1-GFP levels in WT cells transfected with control (siGFP and siScr, grey) or siRNAs targeting FoxO1 (siFoxO1, red). One representative of *n* = 3 independent experiments is shown. **b** Volcano plot showing RNA-seq data of WT cells at N24, treated with siRNA against *FoxO1*. DEGs (*p* adj. ≤ 0.2) that are bound by FoxO1 are colour coded depending on whether they are bound only in 2i (green), only at N24 (blue) or in both conditions (purple). Selected naïve and formative genes are indicated in the plot. For each quadrant, percentages (%) of FoxO1 2i-only, N24-only and 2i&N24 targets are indicated. **c** CUT&RUN analysis of H3K27ac levels on indicated enhancers after FoxO1 siRNA treatment. Signal was normalised to a genomic background region that did not exhibit any H3K27ac signal in WT cells. Data were further normalised to a H3K27ac peak found in *Drosophila* spike-in cells. Data are shown as relative to siScr control. *n* = 2 biological replicates. An independent experiment with further *n* = 2 biological replicates shown in Supplementary Fig. 5g.

To evaluate whether activation of formative-specific enhancers depends on FoxO1 action, we performed CUT&RUN experiments. We assayed H3K27ac levels as a proxy for enhancer activity on a set of known formative gene enhancers in presence and after depletion of *FoxO1*. These experiments showed that indeed activity of enhancers of *Dnmt3b*, *Pou3f1* and *Sall2* and the enhancer of the known FoxO1 target *Enpp2* are dependent on normal *FoxO1* levels (Fig. 5c and Supplementary Fig. 5g).

Together, our results show that proper levels of FoxO1 are required for the effective exit from naïve pluripotency and that loss of FoxO1 abrogates proper formative pluripotency-specific enhancer activation.

**Formative GRN activation by forced nuclear FoxO1 shuttling**
If nuclear shuttling of FoxO TFs is indeed involved in initiating the formative GRN, then artificial inactivation of AKT in the naïve pluripotent state and consequent nuclear accumulation of FoxO TFs should trigger expression of formative-specific genes. To test this hypothesis, we performed RNA-seq after treatment with the AKT inhibitor MK-2206 in 2i and at N24. Consistent with data shown before (Supplementary Fig. 2f), PCA analysis indicated that MK-2206 treatment propelled cells towards a more differentiated state, both when added to naïve ESCs and to cells exiting naïve pluripotency. (Supplementary Fig. 6a). Importantly, FoxO1 targets identified by ChIP-seq represented 27% and 30% of all deregulated genes upon MK-2206 treatment in 2i cells or at N24, respectively (Fig. 6a). Moreover, 44 out of the 105 DEGs in 2i that were not direct FoxO1 targets were differentially expressed during naïve exit in WT cells. MK-2206 treatment in the naïve state led to a highly significant upregulation of FoxO1 N24 targets but had less of an impact on genes that are already FoxO1 targets in 2i (Fig. 6b).

Core formative FoxO1 targets *Pou3f1*, *Pim2*, *Dnmt3b*, *Lef1*, *Fgf5*, *Otx2* and *Sall2* showed a stronger response to MK-2206 treatment in both 2i and at N24 than non-targets (*Hes6*, *Sox12*, *Sox3*, *Tead2*) (Fig. 6c, d). Consistent results were obtained using a larger set of formative-specific genes (Fig. 6e). Interestingly, FoxO1-bound components of the naïve GRN also showed a positive response to MK-2206 treatment in 2i medium (Fig. 6f). A specific response to FoxO signalling rather than mTORC1 deregulation is shown by the fact that treatment with Rapamycin had no specific effect on FoxO1 targets (Supplementary Fig. 6b). Together, this suggests that AKT inhibition and subsequent nuclear FOXO1 translocation is sufficient to initiate major components of the formative GRN, even in ground state culture conditions in 2i medium.

In sum, our work uncovers a role for FoxO TFs downstream of AKT signalling in facilitating the transition from a naïve to a formative pluripotency specific GRN. Our data shows that this is achieved by FoxO1 targeting and regulating large parts of the formative and naïve specific GRNs. Hence, FoxO TFs are pivotal factors in mediating the transition from naïve to formative pluripotency.

## Discussion
In this work, we uncovered a mechanism through which PTEN-mediated AKT regulation controls the transition from naïve to formative pluripotency by regulating nuclear FoxO TF localisation. We demonstrate that FoxO TFs play a fundamental role in orchestrating the exit from naïve pluripotency. Their activity is precisely gated by the activity of PTEN and released once the exit from naïve pluripotency commences (Fig. 7). We propose a mechanism for FoxO1 akin to an actuator in mechanical systems that converts control signals into physical action. Accordingly, FoxO1 acts as an actuator by converting signalling inputs into biological outcomes. By facilitating coordinated dismantling of the naïve and ordered establishment of the formative GRNs, FoxO1 ensures timely and faithful differentiation.

FoxO TFs are well-established regulators of multiple fundamental cellular processes including stress response, DNA repair, cell cycle and apoptosis, metabolism and ageing[45]. Although mostly studied for their role in apoptosis, longevity, and cancer, previous studies have reported a function for FoxO TFs in the regulation of cell fate[38,46–51]. A recent report has shown an important role for FoxO1 in regulating diapause in mouse embryos[52]. Moreover, consistent with a role in establishing formative pluripotent identity, FoxO TFs are reported inhibitors of reprogramming to a naïve pluripotent ESC state[53,54].

FoxO4 was proposed to be the only FoxO TF family member necessary for human ESC differentiation into the neuronal lineage[49], whereas FoxO1 depletion resulted in loss of ESC pluripotency in human and mouse ESCs[51]. Consistent with the latter findings, we could not establish FoxO1 KO ES cells. We cannot exclude that this was due to technical rather than biological hurdles, but note that exclusively in 2i, FoxO1 binds to a number of E3-ligases including *Mdm2*, *Fbx17* and *Huwe1* and REACTOME pathway analysis of 2i-only bound genes shows an enrichment for the term 'Regulation of TP53 Expression and

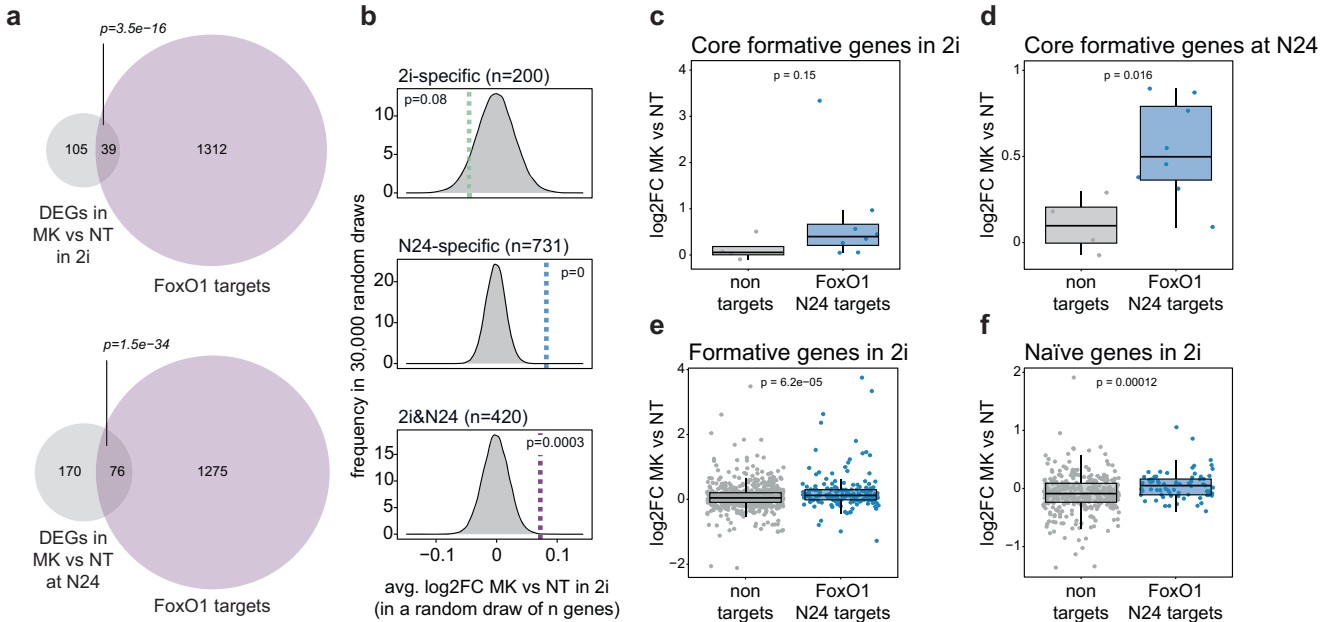

**Fig. 6 | Enforcing nuclear FoxO1 shuttling impairs the transition from the naïve to the formative GRN. a** Venn diagrams showing the overlap between FoxO1 targets (purple) and genes differentially expressed upon MK-2206 treatment in 2i (top) or at N24 (bottom) (DEGs, *p* value ≤ 0.05, grey). *p* values derived from hypergeometric tests of the overlaps are indicated. **b** Bootstrapping analysis (30,000 draws) of gene-sets of the same size as the sample gene sets (2i-specific, 200 genes; N24-specific, 731 genes; 2i&N24, 420 genes). The distribution of average log2FC is plotted. Empirical *p* values for the observed log2FC changes in FoxO1 target gene groups are shown. **c** Box plot showing the expression of core formative genes (*n* = 12) in WT cells after MK-2206 treatment in 2i, divided into FoxO1 N24 targets (blue) and non-targets (grey). Data are shown as log2FC relative to NT cells.

*p* value from two-tailed Wilcoxon rank sum test is indicated. **d** Box plot showing the expression of core formative genes (*n* = 12) in WT cells after MK-2206 treatment at N24, divided into FoxO1 N24 targets (blue) or non-targets (grey). Data are shown as log2FC relative to NT cells. *p* value from two-tailed Wilcoxon rank sum test is indicated. **e** Box plot showing the expression of formative genes (*n* = 814) in WT cells treated with MK-2206 in 2i, divided into FoxO1 N24 targets (blue) or non-targets (grey). Data are shown as log2FC relative to NT cells. *p* value from two-tailed Wilcoxon rank sum test is shown. **f** Box plot showing the expression of naïve genes (*n* = 411) in WT cells treated with MK-2206 in 2i, divided into FoxO1 N24 targets (blue) or non-targets (grey). Data are shown as log2FC relative to NT cells. *p* value from two-tailed Wilcoxon rank sum test is indicated.

Degradation'. This could indicate a possible role for FoxO1 in maintenance of proper protein homoeostasis or cell cycle control in ESCs.

Relocating transcription factors from the cytoplasm to the nucleus or vice versa allows cells to rapidly respond to changes in signalling inputs by providing a cell state switch with rapid on-off kinetics. A relevant example is the regulation of TFE3, a bHLH transcription factor that is relocated from the nucleus to the cytoplasm. TFE3 is found in the nucleus of naïve ESCs, where it sustains the naïve GRN via transcriptional control of *Esrrb*[27]. Induced by a metabolic shift, mTORC1-dependent and mTORC1-independent nutrient-sensing pathways converge to cause TFE3 export from the nucleus, thus contributing to the extinction of the naïve GRN[27,31]. Whether the export of TFE3 and the import of FoxO TFs are coordinated remains an interesting open question.

Once translocated to the nucleus, FoxO TFs play key roles in the establishment of the formative GRN and contribute to the expression of many genes that are themselves required for the exit from naïve pluripotency. Our ChIP-seq experiment revealed that FoxO1 binds to multiple genomic locations even in naïve conditions and that FoxO1 2i targets are part of the core naïve TF-network. Hence, FoxO1 could be necessary for the maintenance of naïve identity by directly regulating the naïve TF-network. However, FoxO TFs remain bound to multiple core naïve genes even 24 h after the onset of differentiation, when most of them have been transcriptionally inactivated. These binding dynamics pose the question of whether FoxO TFs can act not only as activators, but also as repressors depending on cellular context. Such a role is consistent with the large cohort of naïve TFs bound by FoxO TFs during differentiation, the observed downregulation of several naïve genes upon FoxO1 nuclear overexpression and the increased levels of FoxO TF-bound core naïve genes after FoxO1 siRNA treatment.

But how can such a silencing function be reconciled with the fact that FoxO TFs are mainly known as activators of gene expression? In a recent study on human transcriptional effector domains it was shown that FoxO1 can display activator and repressor activity[55]. Hence, whether FoxO TFs act as activators or repressors might depend on cellular environment, post-translational modifications, or on the co-binding of other TFs and co-factors. This potential dual role of FoxO TFs in mediating both activation and silencing will be an exciting question for future research.

Our data show that many FoxO TF-bound genomic regions are enhancers that are also bound by ESRRB, OCT4, NANOG, OTX2, β-CATENIN and TCF7L1. This places FoxO TFs as a core component of both naïve and formative GRNs. β-CATENIN was reported to physically interact with FOXO4[56,57] to increase its activity[43]. Interaction between FoxO TFs and TCF transcription factor derived peptides, in turn, was reported to disrupt β-CATENIN condensate formation, thereby interfering with β-catenin-driven gene expression[58]. Consistent with a repression enhancing role for *FoxO1* in cooperation with *Tcf7l1*, FOXO1-TCF7L1 co-bound genes show a significantly stronger downregulation compared to single factor targets. These results are exciting, because they suggest that AKT regulates Wnt signalling not only through phosphorylation of GSK3 but achieves both positive and negative regulation of Wnt signalling target genes through interaction of FOXO1 with β-CATENIN and TCF7L1, respectively.

It is also tempting to speculate about a function for cytoplasmic phosphorylated FoxO TFs, which are abundant in the naïve state. Cytoplasmic phosphorylated FoxO TFs can bind to the scaffold protein IQGAP1, and this interaction prevents IQGAP1-mediated ERK-activation[59]. Such an interaction would enable cytoplasmic phospho-FoxO to stabilise naïve identity by inhibiting ERK signalling. Nuclear

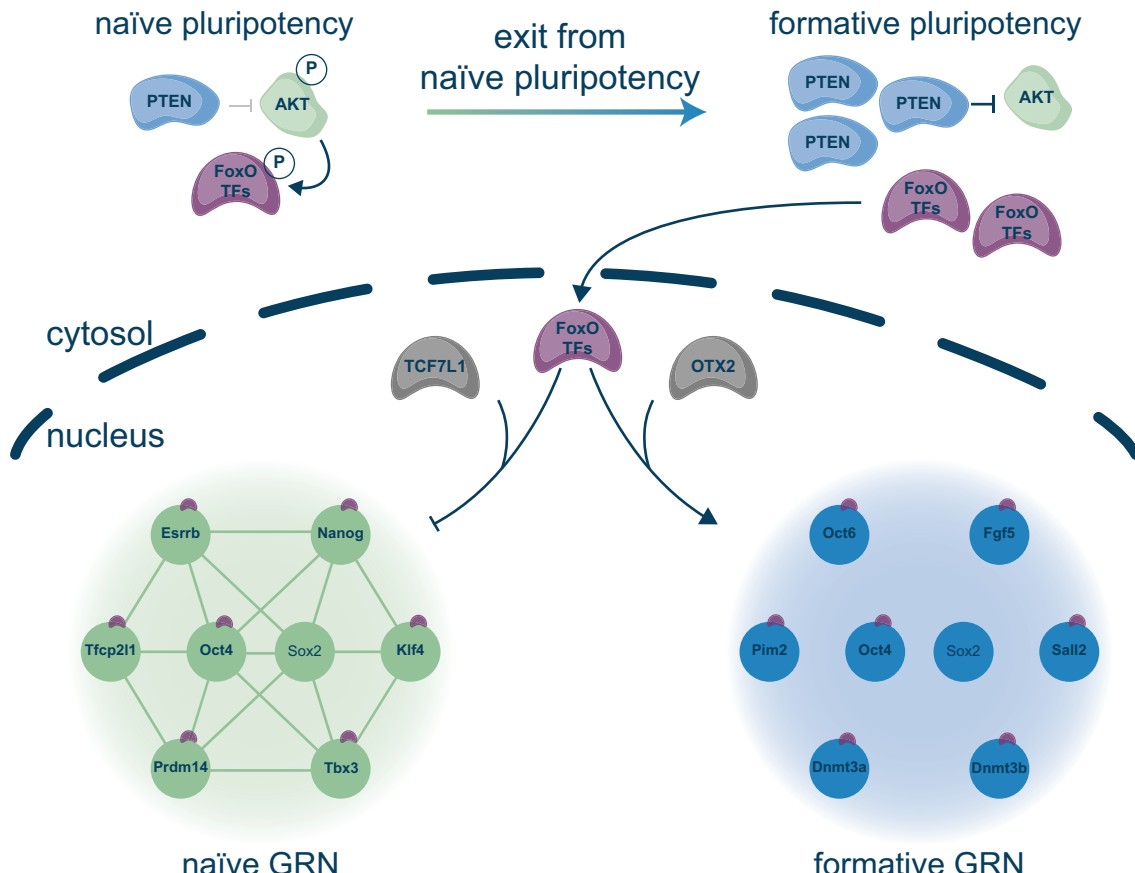

**Fig. 7 | AKT signalling regulation of the naïve to formative pluripotency transition through FoxO TFs.** Schematic illustration of the proposed model of FoxO TF action at the exit from naïve pluripotency.

translocation of FoxO TFs at the onset of differentiation would release ERK inhibition and facilitate the exit from naïve pluripotency. Hence, FoxO nuclear shuttling could execute a dual role to firstly transcriptionally control the naïve and formative core GRNs, and secondly to allow ERK to exert its crucial function at the exit from naïve pluripotency. This would position FoxO TFs as a central cell fate switch that can both shield naïve identity and disrupt it under differentiation-permissive conditions but remains pure speculation.

FoxO TFs are classically seen as tumour suppressors, but a pro-tumorigenic role has been proposed[60,61]. Whether the control of cell fate specific programmes in response to a shift in signalling states, as reported here, contributes to the role of FoxO TFs in tumour biology remains to be investigated.

Our work shows that AKT functionally controls at least two major regulatory axes that regulate the transition to post-implantation-like pluripotency, mTORC1 and FoxO TFs. Our epistasis experiments suggest only a minor role for GSK3, which was reported before[23]. However, in the absence of experimental evidence, we cannot formally exclude a contribution of other targets apart from mTORC1 and FoxO TFs that might contribute to proper cell fate transition to formative identity.

Interestingly, FoxO1 is not only regulating embryonic peri-implantation development, but also acts on the maternal side of the pregnancy as critical regulator of endometrial receptivity in vivo[62,63]. This suggests a concomitant pivotal dual role for FoxO1 at the time of implantation: to initiate post-implantation epiblast fate in the embryo and to prepare the uterus for proper implantation of the same.

In sum, our work highlights a previously unappreciated role of AKT in safeguarding pluripotency by ensuring cytoplasmic sequestration of FoxO TFs. Upon nuclear translocation, FoxO TFs play a key role in the shutdown of the naïve pluripotent identity and the initiation of the formative gene expression programme. Our findings place FoxO transcription factors, and specifically FoxO1, as a crucial component of the pluripotency TF-network with a pivotal role to instruct formative fate. Signalling-instructed translocation of lineage determining TFs such as FoxO1 provides a paradigm that allows the precise and effective control of delicately balanced GRNs that govern stem cell transitions.

## Methods
### Cell culture
Mouse embryonic stem cells (mESCs) were cultured on gelatin-coated (Sigma-Aldrich, G1890) plates in DMEM high-glucose medium (Sigma-Aldrich, D5671) supplemented with 10% FBS (Gibco, 10270-106), 2 mM L-Glutamine (Sigma-Aldrich, G7513), 0.1 mM NEAA (Sigma-Aldrich, M7145), 1 mM Sodium Pyruvate (Sigma-Aldrich, S8636), 10 μg/ml penicillin-streptomycin (Sigma-Aldrich, P4333), 55 μM β-mercaptoethanol (Fisher Scientific, 21985-023), 10 ng/ml LIF (batch tested, in-house) and 2i (1.5 μM PD0325901 and 3 μM CHIR99021), referred to here as ES DMEM-2i medium. mESCs were passaged every second day and routinely tested negative for mycoplasma infection[28,64].

All cell lines used in this study were derived from a parental cell line carrying a Rex1-GFPd2 reporter (destabilised version of the GFP transgene (GFPd2) under control of the endogenous Rex1 promoter[22]) and a Cas9 transgene (EF1alpha-Cas9 cassette targeted to the Rosa26 locus[30]) (RC9 cells). Cells lacking *Pten*, *Tsc2* or *Tcf7l1* (*Pten* KO, *Tsc2* KO and *Tcf7l1* KO) had been generated previously[28]. Pten-rescue cell lines were generated by cloning the *Pten* coding sequence (amplified by PCR from RC9 genomic DNA) into a pCAG-3xFLAG-empty-pgk-hph vector[27].

All work reported here complies with all relevant ethical regulations applicable to research conducted at the Max Perutz Laboratories and the University of Vienna. All animal experiments were approved by the IMP/IMBA animal house and performed in accordance with institutional guidelines.

## Monolayer differentiation

mESCs were plated on gelatin-coated plates at a final density of $1 \times 10^4$ cells/cm$^2$ in N2B27 medium (1:1 mix of DMEM/F12 (Gibco, 21331020) and Neurobasal medium (Gibco, 21103049) supplemented with N2 (homemade), B27 Serum-Free Supplement (Gibco, 17504-044), 2 mM L-Glutamine (Sigma-Aldrich, G7513), 0.1 mM NEAA (Sigma-Aldrich, M7145), 10 µg/ml penicillin-streptomycin (Sigma-Aldrich, P4333), 55 µM β-mercaptoethanol (Fisher Scientific, 21985-023)) and 2i (1 µM PD0325901 and 3 µM CHIR99021), here referred to as N2B27-2i medium. The following day, cells were washed with PBS and medium was exchanged to either N2B27 without 2i to induce differentiation for the indicated time, or to fresh N2B27-2i for undifferentiated controls.

## Rapamycin treatment

mESCs were plated on gelatin-coated plates in N2B27-2i + DMSO (Sigma-Aldrich, D2650) or N2B27-2i + 20 nM Rapamycin (Enzo Life Sciences, BML-A275-0005). The following day, cells were washed with PBS and medium was exchanged to N2B27 + DMSO or N2B27 + 20 nM Rapamycin to induce differentiation for the indicated time.

## MK-2206 treatment

mESCs were plated on gelatin-coated plates in N2B27-2i or N2B27-2i + 1 µM MK-2206 (Cayman Chemical, 11593). The following day, cells were washed with PBS and medium was exchanged to either N2B27 or N2B27 + 1 µM MK-2206 to induce differentiation for the indicated time, or to N2B27-2i or N2B27-2i + 1 µM MK-2206 for the undifferentiated controls.

## RNAi assay

mESCs were plated on gelatin-coated plates in N2B27-2i and transfected with FlexiTube siRNAs (Qiagen) using DharmaFECT 1 (Fisher Scientific, T-2001). The next day, cells were washed with PBS and medium was exchanged to either N2B27 without 2i to induce differentiation, or fresh N2B27-2i for the undifferentiated controls. Two siRNAs targeting FoxO1 (SI01005200, SI02694153) were used at a final concentration of 20 nM. As controls, siRNAs targeting GFP (siGFP) or scrambled siRNAs (siScr) were used.

## Expression of nuclear FoxO1

A FoxO1 coding sequence carrying three mutations (T24A, S253D and S316A) was cloned from an Addgene vector (Plasmid #12149)[65] into a pB-TetOn-3xFLAG-Empty-PolyA-Puro vector. The resulting plasmid, here referred to as 3xFLAG-FoxO1[nuc], was transfected into RC9 and *Pten* KO cells. Single cell derived, independent clones were selected and expanded for further experiments. For experiments in differentiation-permissive conditions, RC9-based and *Pten* KO-based clones were plated on gelatin-coated plates in N2B27-2i. The following day, cells were washed with PBS and medium was exchanged to N2B27 with or without 500 ng/ml of doxycycline (Sigma-Aldrich, D9891). After 8 h, medium was changed to N2B27, and cells were differentiated for further 16 h. For the experiments in naïve conditions, RC9-based clones were plated in N2B27-2i and cultured for 48 h. The last 8 h before harvesting, 500 ng/ml of doxycycline was added to the medium.

## Flow cytometry analysis

After the indicated amount of time in differentiation (N2B27-based) or control (N2B27-2i-based) medium conditions, cells were harvested using 0.25% trypsin/EDTA and resuspended in ES-DMEM medium to neutralise trypsin. Rex1-GFPd2 levels were measured using the LSR Fortessa flow cytometer (BD bioscience) and then analysed with the FlowJo software (v10, BD bioscience).

## Real-time PCR analysis

After the indicated amount of time in differentiation (N2B27-based) or control (N2B27-2i-based) medium conditions, cells were washed with PBS and harvested in RNA Lysis buffer containing 1% (v/v) 2-mercaptoethanol and stored at −80 °C before isolation of RNA. RNA was extracted using the ExtractMe kit (Blirt, EM15) following the manufacturer's instructions. 0.25–1 µg of RNA was reverse-transcribed into cDNA using the SensiFAST cDNA Synthesis Kit (Bioline, BIO-65054). Real-time PCR was performed on the CFX384 Touch real-time PCR detection system (Bio-Rad) using the Sensifast SYBR no Rox kit (Bioline, BIO-98020). Data analysis and visualisation was performed using Microsoft Excel (Office 365) and R (v4.2.2). Used primers are listed in Supplementary Data 3.

## Immunoblotting analysis

After the indicated amount of time in differentiation (N2B27-based) or control (N2B27-2i-based) medium conditions, cells were washed with PBS and harvested in 1x RIPA buffer (Sigma-Aldrich, 20-188) supplemented with Complete Mini EDTA-free protease inhibitor cocktail (Roche, 04693159001) and PhosSTOP (phosphatase inhibitor cocktail (Roche, 04906845001). Protein extraction was performed by incubating the samples on ice for 15 min and then collecting the supernatant after centrifugation at $14,000 \times g$ at 4 °C for 30 min. Protein concentration was determined using a Bradford Assay (Bio-Rad). Eight to twenty µg whole cell lysates were separated on 8–12% SDS–PAGE gels (depending on the molecular weight of the target proteins) and subsequently blotted on 0.2 µm nitrocellulose membranes (Amersham). Membranes were blocked at RT for 1 h with 5% milk diluted in PBS (Sigma-Aldrich, P4417) containing 0.1% Tween-20 (PBS-T). Primary antibodies were incubated overnight at 4 °C or for 1 h at room temperature (RT). Secondary antibodies were incubated for 1 h at RT. The following primary antibodies were diluted in PBS-T containing 5% BSA and used 1:1000 for anti-phospho-Akt(Ser473) (rabbit; Cell Signalling, 4058), 1:1000 for anti-phospho-Akt(Thr308) (rabbit; Cell Signalling, 13038), 1:1000 for anti-pan-Akt (rabbit; Cell Signalling, 4691), 1:1000 for anti-phospho-GSK3β(Ser9) (rabbit; Cell Signalling, 9336), 1:1000 for anti-GSK3β (rabbit; Cell Signalling, 12456), 1:1000 for anti-phospho-4E-BP1(Ser65) (rabbit; Cell Signalling, 9451), 1:1000 for anti-4E-BP1 (rabbit; Cell Signalling, 9644), 1:1000 for anti-phospho-p70 S6 kinase(Thr389) (rabbit; Cell Signalling, 9234), 1:1000 for anti-p70 S6 kinase (rabbit; Cell Signalling, 2708), 1:1000 for anti-PTEN (rabbit; Cell Signalling, 9559), 1:1000 for anti-TSC2 (rabbit; Cell Signalling, 4308), 1:1000 for anti-FoxO1 (rabbit; Cell Signalling, 2880), 1:1000 for anti-phospho-FoxO1(Ser256) (rabbit; Cell Signalling, 9461) and 1:1000 for anti-FoxO3a (rabbit; Cell Signalling, 2497). The following primary antibodies were diluted in PBS-T containing 5% milk and used 1:1000 for anti-Oct4 (rabbit; Abcam, ab19857), 1:5000 for anti-Tubulin (mouse; Sigma-Aldrich, T8203), 1:10,000 for anti-GAPDH(G-9) (mouse; Santa Cruz Biotechnology, sc-365062) and 1:10,000 for anti-Vinculin(H-10) (mouse; Santa Cruz Biotechnology, sc-25336). Secondary antibodies were diluted in 5% milk and used 1:10,000 for anti-rabbit IgG (Amersham, NA934) and 1:15,000 for anti-mouse IgG (goat; Santa Cruz Biotechnology, sc-2064). Chemiluminescence signal was detected using the ECL Select detection kit (GE Healthcare, GERPN2235) with a ChemiDoc system (Bio-Rad). Data analysis was performed using ImageLab (v5.2.1). Uncropped images are available in the Source Data File.

## Immunofluorescence analysis

mESCs were plated at a final density of $1 \times 10^4$ cells/cm$^2$ on fibronectin-coated (MM Merck Millipore, FC010) µ-Slide 8 Well Glass Bottom Chamber Slides (Ibidi, 80827) in N2B27-2i medium. The following day,

cells were washed with PBS and medium was exchanged to either N2B27 without 2i to induce differentiation, or to fresh N2B27-2i for the undifferentiated controls. After the indicated amount of time, cells were washed with PBS and fixed for 15 min at RT with freshly diluted 4% PFA (16% paraformaldehyde diluted in 1:4 in PBS) (SCI Science Services, E15710). Cells were washed in PBS and subsequently permeabilized with PBS containing 0.1% Triton-X for 10 min at RT. Cells were washed 3x in PBS-T and then blocked using PBS-T containing 5% BSA (blocking buffer) for 30 min at 4 °C. Primary antibodies were incubated overnight at 4 °C. Cells were washed 3x with PBS-T. Secondary antibodies were incubated for 1 h at RT. Cells were washed 2x with PBS-T. Nuclei were stained with 1 μg/ml DAPI (Sigma-Aldrich, D9542) for 10 min at RT. Cells were washed 3x with PBS and stored in PBS at 4 °C until the image acquisition procedure. Primary and secondary antibodies were diluted in blocking buffer and used 1:100 for anti-FoxO1 (rabbit; Cell Signalling, 2880), 1:100 for anti-ESRRB (mouse; R&D Systems, PP-H6705-00), 1:250 for anti-FLAG M2 (mouse; Sigma-Aldrich F1804), 1:200 for anti-NANOG (rabbit; NovusBio, NB100-58842), 1:500 for anti-mouse Alexa-555 (donkey; Cell Signalling, 4409), 1:500 for anti-rabbit Alexa-647 (goat; Cell Signalling, 4414). Images were acquired using a Zeiss LSM 980 confocal microscope with a Plan-Apochromat 63x/1.4 Oil DIC M27 (WD 0.19 mm) objective. Images were analysed using Fiji/ImageJ (v2.9.0/1.54f). For quantifying the nuclear intensity of the stained proteins, segmentation was performed on the DAPI channel with cellpose with the following settings: cell diameter (in pixels) = 170, flow_threshold = 0.4, cellprob_threshold = 0.0, stitch_threshold = 0.0, model = cyto2. The obtained nuclei outlines were imported into Fiji/ImageJ and used to create mask objects. The latter were used to measure the mean fluorescence intensity of each nucleus in all recorded channels. Nuclei out of focus were manually removed from the analysis. Data analysis and visualisation was subsequently performed in R.

### Intracellular staining

After the indicated amount of time in differentiation (N2B27-based) or control (N2B27-2i-based) medium conditions, cells were harvested using 0.25% trypsin/EDTA and resuspended in ES-DMEM medium to neutralise trypsin. Cells were centrifuged, and cell pellets were washed twice with PBS before fixation with 2% PFA for 15 min at RT. Cells were washed with FACS buffer (PBS containing 5% BSA) and subsequently permeabilized with ice-cold MeOH for 10 min on ice. After 3 washes in FACS buffer, cells were incubated in FACS buffer for 10 min in the dark. Subsequently, cells were incubated with primary antibodies for 1 h at RT. Cells were then washed 3x with FACS buffer and incubated with secondary antibodies for 15 min on ice. Cells were washed 3x with FACS buffer and stored in FACS buffer until flow cytometry analysis at the LSR Fortessa flow cytometer. Primary and secondary antibodies were diluted in FACS buffer and used 1:100 for anti-phospho-Akt(Ser473) (rabbit; Cell Signalling, 4058) and 1:500 for anti-rabbit Alexa-647 (goat; Cell Signalling, 4414). Flow cytometry data was analysed with the FlowJo software. Mean fluorescence intensities (MFI) for stained samples were calculated by subtracting the MFI of their relative controls (cells stained only with the secondary antibody). Data analysis and visualisation was performed in R.

### Embryo staining

Blastocysts from B6CBAF1 mice were isolated at embryonic stages E3.5, E4.5, E4.75 and E5.5. The samples were fixed with 4% paraformaldehyde for 30 min and stored in 1X PBS at 4 °C. To prepare samples for IF, they were permeabilized with 0.3% PBST (PBS + Triton) for 60 min at room temperature before incubating with Blocking buffer (10% Donkey Serum, in 0.3% PBST) for 3 h. Primary antibody incubation was performed overnight at 4 °C in blocking buffer. All primary antibodies were used at a dilution of 1:200. Prior adding the secondary antibody, samples were washed three times with 0.3% PBST.

The embryos were incubated with the secondary antibody for 1 h at room temperature before washing three times with 0.3 % PBST. All the secondary antibodies and Hoechst staining were used at a dilution of 1:300. The samples were then stored at 4 °C in PBS prior imaging. The following antibodies were used: Gata4 (Monoclonal, Mouse, Santa Cruz Biotechnology, sc-25310) + Secondary Ab (Anti-Mouse, Donkey, AF 647, Invitrogen A31571), Cdx2 (Monoclonal, Mouse, Emergo Europe/BioGenex MU392A-5UC) + Secondary Ab (Anti-Mouse, Donkey, AF 647, Invitrogen A31571), FoxO1 (rabbit C29H4) + Secondary Ab (Anti-Rabbit IgG, Donkey, AF 568, Invitrogen A10042), Sox2 (Monoclonal, Rat Invitrogen 14-9811-82) + Secondary Ab (Anti-Rat, Donkey, AF 488, Invitrogen A21208), Hoechst (Life Technology Corporation, 33342). Images were obtained on an Olympus spinning disk confocal (inverted) with 40x/0.75 (Air) WD 0.5 mm objective. Images were analysed using Fiji/ImageJ (v2.0.0/1.53t). For each embryonic stage, at least 3 embryos were analysed. Single nuclei were manually segmented at multiple z-sections, and the mean intensity was calculated using the Fiji Measure tool.

### Nucleo-cytoplasmic fractionation

Subcellular fractionation experiments were performed following a protocol adapted from Rockland (https://www.rockland.com/resources/nuclear-and-cytoplasmatic-extract-protocol/). A total of $1 \times 10^4$ cells/cm² of WT and *Pten* KO cells were plated in 10 cm gelatine-coated plates. The next day, cells were washed with PBS and medium was exchanged to either N2B27 without 2i to induce differentiation, or to fresh N2B27-2i for the undifferentiated controls. After 24 h cells were harvested. 1/20 of cells were processed as described above for total protein extraction. The rest was used for the nucleo-cytoplasmic fractionation. Cell pellets were resuspended in 5 pellet volumes of CE buffer adjusted to pH 7.6 (10 mM HEPES, 60 mM KCl, 1 mM EDTA, 0.075% NP-40, 1 mM DTT, 1 mM PMSF) and incubated on ice for 3 min. Samples were pelleted by centrifugation at $300 \times g$ for 4 min at 4 °C and the supernatant (the cytoplasmic fraction) was transferred to clean tubes. The nuclei were washed carefully with 5 pellet volumes of CE buffer (without NP-40), and then pelleted at $300 \times g$ for 4 min at 4 °C. The supernatant was discarded, and the nuclei were resuspended in 1 pellet volume of NE buffer adjusted to pH 8.0 (20 mM Tris Cl, 420 mM NaCl, 1.5 mM MgCl2, 0.2 mM EDTA, 1 mM PMSF, 25% glycerol) and the salt concentration was then adjusted to 400 mM with 5 M NaCl. An additional pellet volume of NE buffer was added to the extracts before incubating them for 10 min on ice. The extracts were vortexed every 3 min during the incubation time. Both cytoplasmic and nuclear fractions were spun at maximum speed to pellet any remaining nuclei. Cytoplasmic and nuclear fractions were transferred to clean tubes; 20% glycerol was added to the cytoplasmic fraction. Both fractions were stored at −80 °C.

### RNA-sequencing analysis

For differentiation experiments using RC9, *Pten* KO and *Tsc2* KO cells, and for the 2h-resolved time course, count tables generated in a previous study were used[28]. QuantSeq analysis was performed for all the other RNA-seq experiments described in the manuscript. For the Rapamycin experiment, WT, *Pten* KO and *Tsc2* KO cells treated with DMSO or Rapamycin from two independent differentiation assays were sequenced (duplicates). For the FoxO1 knockdown experiment, WT cells were treated either with an individual siRNA sequence targeting *FoxO1* transcript, or with a combination of both siRNAs (triplicates, siRNA #1, siRNA#2, siRNA#1 + siRNA#2) in a reverse transfection approach[64]. For the control sample (siCtrl), two siGFP samples combined with 1 siScr sample were considered a triplicate. For the MK-2206 experiment, WT cells left untreated or treated with MK-2206 from two independent differentiation assays were sequenced (duplicates). Library preparation (according to the Lexogen 3' mRNA Seq Library Prep Kit), multiplexing (by qPCRs) and

sequencing on an Illumina NextSeq2000 P3 platform was carried out at the VBCF NGS facility. Five to ten million of single-end reads at 50 bp read length were generated per sample. The resulting fastq files were analysed with a Nextflow 23.04.1.5866/nf-core/rnaseq v3.10.1 pipeline. Quality control was performed using fastQC (v0.11.9), and transcripts were mapped to the mm10 assembly mouse reference genome using Salmon (v1.9.0) as a pseudoaligner and STAR (v2.7.10a) as an aligner. DESeq2 (v1.38.3) was used to generate normalised count tables and to perform differential expression analyses (FDR-adjusted $p$ value ≤ 0.05; H0: log2FC = 0). pheatmap (v1.0.12), EnhancedVolcano (v1.16.0), UpsetR (v1.4.0), eulerr (v7.0.0) and ggplot2 (v3.4.0) were used for data visualisation in R. Combined lists of upregulated and downregulated genes in *Pten* KO and *Tsc2* KO were generated by selecting genes differentially expressed (DEGs) in both KOs (log2FC ≥ 0.5 for the upregulated genes, and log2FC ≤ −0.5 for the downregulated genes). Whenever we compared gene groups from published transcriptomics datasets, we only considered genes that are present also in our datasets.

## Chromatin immunoprecipitation (ChIP)

FoxO1 and FoxO3 ChIP were performed as described[41]. $1.5 \times 10^4$ cells/cm² of WT and *Pten* KO cells were plated in duplicate (for *Pten* KO cells, each replicate corresponded to a different clone) on 15 cm gelatine-coated plates. The next day, cells were washed with PBS and medium was exchanged to either N2B27 without 2i to induce differentiation, or to fresh N2B27-2i for the undifferentiated controls. After 24 h, cells were harvested. Cells were washed with PBS, and then cross-linked directly on the plate with 1% formaldehyde in PBS for 10 min. Subsequently, 0.125 M glycine was added to the plates for 10 min to quench cross-linking. The plates were washed 2x with PBS, and then cells were scraped off in ice-cold PBS containing 0.01% Triton-X. Cells were pelleted by centrifugation at $500 \times g$ for 5 min and flash-frozen in liquid nitrogen. Cell pellets were resuspended in 5 ml LB1 (50 mM HEPES pH 7.5, 140 mM NaCl, 1 mM EDTA, 10% glycerol, 0.5% NP-40, 0.25% TX-100, 1 mM PMSF, 1 × Complete Mini EDTA-free protease inhibitor cocktail) to extract nuclei and rotated vertically for 10 min at 4 °C. Nuclei were pelleted by centrifugation at $1350 \times g$ for 5 min at 4 °C, and then resuspended in 5 ml LB2 (10 mM Tris pH 8.0, 200 mM NaCl, 1 mM EDTA, 0.5 mM EGTA1, mM PMSF, 1xComplete Mini EDTA-free protease inhibitor cocktail), and rotated vertically for 10 min at RT. Samples were pelleted by centrifugation at $1350 \times g$ for 5 min at 4 °C and then resuspended in 1.5 ml LB3 (10 mM Tris–HCl pH 8.0, 100 mM NaCl, 1 mM EDTA 0.5 mM EGTA, 0.1% Na-deoxycholate, 0.5% N-lauroylsarcosine, 1 mM PMSF, 1xComplete Mini EDTA-free protease inhibitor cocktail) and 200 µl sonication beads (diagenode) in Bioruptor® Pico Tubes (diagenode). Chromatin was sonicated for 13 cycles with 30 s on, 45 s off parameters. Sonicated samples were transferred to fresh tubes and centrifuged at $16,000 \times g$ at 4 °C to pellet cellular debris. 1.1 ml of supernatant were collected and transferred to fresh tubes. A total of 110 µl of 10% Triton-X were added to a final concentration of 1%. For each sample, 50 µl were collected as input and 1 ml was used for immunoprecipitation. Chromatin was incubated with 20 µl (1:50 dilution) of anti-FoxO1 (rabbit; Cell Signalling, 2880) or anti-FoxO3a (rabbit; Cell Signalling, 2497) antibodies overnight at 4 °C with vertical rotation. Samples were collected after 14 h. A total of 100 µl of Dynabeads protein G (Thermo Fisher Scientific, 10765583) per sample were washed in ice-cold blocking solution (PBS containing 0.5% BSA) and then incubated with the antibody-bound chromatin solutions for 4 h. Beads were washed 5x in ice-cold RIPA wash buffer (50 mM HEPES pH 7.5, 500 mM LiCl, 1 mM EDTA, 1% NP-40, 0.7% Na-Deoxycholate), and then 3x with TE + 50 mM NaCl. Samples were eluted in 210 µl elution buffer (50 mM Tris pH 8.0, 10 mM EDTA, 1% SDS) for 15 min at 65 °C. Supernatant containing the antibody-bound chromatin fraction was separated from the beads. Three volumes of elution buffer were added to the input samples. Decrosslinking was performed by incubating both input and ChIP samples at 65 °C overnight. The next day, one volume of TE and RNaseA (to a 0.2 mg/ml final concentration) were added to the input and ChIP samples, followed by incubation for 2 h at 37 °C. Final salt concentration was adjusted to 5.25 mM $CaCl_2$ with 300 mM $CaCl_2$ in 10 mM Tris pH 8.0. Samples were then incubated with 0.2 mg/ml Proteinase K for 30 min at 55 °C. DNA was extracted with phenol-chloroform using Phase Lock GelTM tubes (Quantabio, 733-2478) and then precipitated in EtOH. DNA pellets were dissolved in $H_2O$.

## ChIP-sequencing analysis

Libraries were prepared at the VBCF NGS facility and sequenced on an Illumina NovaSeq platform. Within the FoxO1 ChIP experiment, 20–40 million paired-end reads at 150 bp read length were generated for the input samples, and 80–130 million reads for the ChIP samples. Within the FoxO3 ChIP experiment, ~ 25 million single-end reads at 100 bp read length were generated for both input and ChIP samples. RC9-based FoxO3 ChIP samples were re-sequenced to increase sequencing depth, and an additional ~ 60 million of paired-end reads at 150 bp read length per sample were generated. R1 reads from the two sequencing runs were concatenated before data processing. FoxO1 ChIP and input reads were processed using a paired-end mode, while FoxO3 ChIP and input samples following a single-end mode. Quality control of fastq files was performed using fastQC (v0.11.9) before and after trimming sequencing adaptor fragments with trim-galore (v0.6.7) and cutadapt (v3.5). Additional 2 bp at the 3' end were removed with the parameters *--three_prime_clip_R1 2* (*--three_prime_clip_R2 2* in case of paired-end reads). The trimmed reads were aligned to the mm10 assembly mouse reference genome with bowtie2 (v2.4.4), with an alignment rate of 80–98%. The obtained sam files were converted to bam files with samtools (v1.13), and uniquely mapping reads were extracted by removing duplicate reads (as potential PCR artefacts) with samtools *markdup*. Peak calling was performed on bam files with macs2 (v2.2.7.1) on combined ChIP duplicates, using all input samples as control files. Potential artefactual regions listed in the mm10 blacklist (http://mitra.stanford.edu/kundaje/akundaje/release/blacklists/mm10-mouse/mm10.blacklist.bed.gz) were removed from the obtained bed files using bedtools (v2.31.0). Peaks assigned to unidentified regions (chrUn) were manually removed.

## ATAC-sequencing

A total of $1 \times 10^4$ cells/cm² cells were plated on gelatine-coated plates. The next day, cells were washed with PBS and medium was exchanged to either N2B27 without 2i to induce differentiation, or to fresh N2B27-2i for the undifferentiated controls. After 24 h cells were harvested and counted. 250,000 cells per sample were submitted to the VBCF NGS facility for further processing and ATAC-seq library preparation (Bulk ATAC-seq Illumina). In brief, cells were lysed with 0.5x lysis buffer (0.01 M Tris-HCl pH 7.5, 0.01 M NaCl, 0.003 M $MgCl_2$, 1% BSA, 0.1% Tween-20, 0.05% NP-40, 0.005% Digitonin, 0.001 M DTT, 1 U/µl RNAse inhibitor), and tagmentation reaction (Tn5 Illumina) was performed on 50,000 isolated nuclei. Libraries were prepared with the Nextera DNA Library Preparation kit and sequenced on an Illumina NovaSeq platform. A total of 80–170 million of paired-end reads at 150 bp read length were generated per sample. The resulting fastq files were analysed with a Nextflow 23.04.1.5866/nf-core/atacseq v2.0 pipeline. Quality control was performed using fastQC (v0.11.9), and reads were mapped to the mm10 assembly mouse reference genome using bowtie2 (v2.4.4). Peak calling was performed on the bam files generated by the nfcore pipeline with Genrich (v0.6.1). Consensus peaks were defined as peaks with at least a 50% overlap between replicates and generated with samtools *intersect*. Bed files were filtered for the mm10 blacklist using bedtools. Peaks assigned to unidentified regions (chrUn) were manually removed.

## CUT&RUN analysis

The ChIC/CUT&RUN Assay Kit (Active Motif, catalogue No. 53180) was used for the CUT&RUN experiment. FoxO1 siRNA treatment was performed as described before using siRNAs directed against FoxO1 and scrambled siRNAs as controls. A total of 125,000 to 500,000 cells were collected per condition in biological duplicates from two independent experiments ($n = 4$). As spike-in, 25,000 *Drosophila melanogaster* S2 cells were added to each sample. CUT&RUN was performed according to the manufacturer's protocol, but ethanol precipitation performed to purify DNA instead of column purification. One μg of Histone H3K27ac antibody (rabbit pAb, Thermo Fisher Scientific, 39133) was used per sample. IgG antibody supplied with the CUT&RUN kit was used as a control. DNA was eluted in 100 μL and used as qPCR template. FoxO1 bound H3K27ac positive enhancer sequences near formative genes and control regions not bound by either FOXO1 or H3K27ac were amplified to assay H3K27ac levels. A known H3K27ac decorated region in *Drosophila* S2 cells (chr3L:4104063-4104149, Aug. 2014- BDGP Release 6 + ISO1 MT/dm6), was used to normalise signal between samples. The qPCR primers used are listed in the Supplementary Data 3.

## Motif enrichment analyses

Motif enrichment analysis was performed with Homer (v4.11). For finding motifs enriched in FoxO1 or FoxO3-bound regions, *findMotifsGenome.pl* was used with default settings. For finding enriched motifs in ESCs or EpiLCs enhancers, enhancer lists from the Bücker lab were used (filtered to exclude TSS[41]). A list of overlapping enhancers between ESCs and EpiLCs was generated with bedtools and used as background to identify ESC-specific and EpiLC-specific enriched motifs with *findMotifsGenome.pl* using default settings. According to the Homer documentation, a motif can be considered robustly enriched when its associated *p* value is lower than $1 \times 10^{-50}$.

## Data integration analyses

Downstream analyses for ChIP-seq and ATAC-seq were performed with Deeptools (v3.5.1). BigWigs were generated from single bam files with *bamCoverage* using a binsize of 10 bp and a normalisation coverage to 1x mouse genome size (RPGC) excluding the X chromosome. Principal component analysis (PCA) was performed with *multiBigwigSummary* and *plotPCA*. Bigwig replicates were merged using *bigWigMerge*, and heatmaps were plotted on merged BigWigs using *computeMatrix* and *plotHeatmap*. Heatmaps were sorted based on FoxO1 or FoxO3 signals. Three lists of peaks (separately for FoxO1 and FoxO3 ChIP) were generated with bedtools by intersecting WT (RC9) bed files: 2i-only peaks, N24-only peaks and shared peaks. Peak to gene association was performed on these peak lists in R with ChIPSeeker (v1.34.1). Oct4 and Otx2 ChIP BigWigs and peak lists were obtained from ref. 39. Esrrb ChIP BigWigs and bed files were obtained from published data[33]. β-catenin ChIP bed files[66] were downloaded from CODEX (https://codex.stemcells.cam.ac.uk/), and mm9 coordinates were converted into mm10 coordinates using the liftOver tool from UCSC (https://genome.ucsc.edu/cgi-bin/hgLiftOver). Overlap between genomic regions was performed with ChIPPeakAnno (v3.32.0). For the FoxO1-FoxO3 ChIP overlap, the background for the hypergeometric test was set to the total number of detected open chromatin regions (generated by merging ATAC-seq peaks in 2i with ATAC-seq peaks at N24 with ESC). For all other overlaps, the background used was obtained by merging all ATAC-seq peaks with the lists of ESC and EpiLC enhancers[41]. Genome tracks shown in Fig. 4 were generated using karyoploteR (v1.24.0) in R. Lists of NANOG, OCT4, SOX2 and TCF7L1 targets were obtained from ref. 67. Gene expression changes during peri-implantation development in vivo were assessed using previously described and processed data[68,28].

## Enrichment analyses

All enrichment analyses presented in this work were performed in R with hypeR (v2.0.1). For KEGG pathway enrichment analysis, mouse KEGG database was downloaded from http://rest.kegg.jp/link/mmu/pathway and transformed into a hypeR-compatible gene-set using the gsets function. The mouse REACTOME dataset was downloaded using the msigdb_gsets function within hypeR. As background, for FoxO1 target enrichment analyses a list of genes associated to any accessible region as detected by ATAC-seq was used, while for the other enrichment analyses the list of all detected genes in the relative RNA-seq experiment was used. Custom gene-sets were also generated with the gsets function. As background for FoxO TF-based enrichments, a list of genes associated with all open chromatin regions was used. Enrichment results were visualised with ggplot2.

## Statistical analysis and data representation

All statistical analyses were performed in R with the ggpubr (v0.6.0) package. Information on statistical tests and replicate numbers are provided in the figure legends. Wherever necessary, correction for multiple testing was performed. All box plots show the 25th percentile (Q1), median (Q2) and 75th percentile (Q3); whiskers indicate minimum (Q1 − 1.5 * inter quartile range [IQR]) and maximum (Q3 + 1.5*IQR) values.

## Reporting summary

Further information on research design is available in the Nature Portfolio Reporting Summary linked to this article.

## Data availability

The NGS data generated in this study have been deposited in the GEO database under accession code GSE253480. Data from the following accession numbers were used in this study: GSE56138; GSE184137; GSE43565. All other relevant data supporting the key findings of this study are available within the article and its Supplementary Information files. Source data are provided with this paper.

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

## Acknowledgements

We thank Kitti Dóra Csályi and Thomas Sauer at the Max Perutz Laboratories FACS Facility for expert support. Next generation sequencing was performed at the Vienna Biocenter Core Facilities (VBCF), and we thank Thomas Grentzinger (VBCF) for ATAC-seq sample processing. We thank Niklas Urbanek, Michael Bindl, Mattia Pitasi, Julia Ramesmayer, Ana Maria Sandru and Stefania Bucconi for experimental support and Ivan Yudushkin for sharing reagents. We thank all members of the Leeb and Buecker labs for helpful discussions and Christa Buecker, Manuela Baccarini, Alexander Stark and Thomas Leonard for support and helpful suggestions. This research was funded in whole, or in part, by the Austrian Science Fund (FWF; dois: 10.55776/I5958 and 10.55776/P35637). L.S. is an OEAW doc.fellowship awardee (DOC/25622) and a member of the FWF-funded doctoral programme 'Signalling Molecules in Cellular Homeostasis' (SMICH; W1261). M.L. is a faculty member and speaker of SMICH and a 'Wiener Wissenschafts- Forschungs- und Technologiefonds (WWTF)' Vienna Research Group Leader (VRG14-006).

## Author contributions

L.S. designed and performed wet-lab and computational experiments and wrote the manuscript. L.M.C.A. and M.H. performed experiments and CUT&RUN analyses. S.K. performed siRNA experiments. A.P.F. performed some image analysis. G.S. and N.R. performed embryo IF stainings and analysis. M.L. designed and supervised the study and wrote the manuscript.

## Competing interests

The authors declare no competing interests.
