## [Peer Review File · Nature Communications]

FoxO transcription factors actuate the formative pluripotency specific gene expression programmeREVIEWER COMMENTS

Reviewer #1 (Remarks to the Author):

Naïve and formative pluripotent stem cells that recapitulate the pre- and early post-implantation epiblast, respectively, have distinct epigenetic and gene expression features. Although the core transcriptional networks that maintain these different pluripotent states have been identified, molecular mechanisms control the transition remain unclear. In this manuscript, the authors address this question using Pten-KO mESCs that reveal a delayed exit of naïve pluripotency upon differentiation and find that FoxO TFs function to initiate the transition from naïve to formative pluripotency by activating the expression of formative pluripotency-specific genes. They utilized ChIP-seq and RNA-seq techniques to identify the target genes of FoxO1 during the exit of naïve pluripotency. The study is very interesting. However, the authors seem to mainly focus on the binding sites of FoxO1 rather than the functional impact of the genes bound by FoxO1 except a small number of core GRN TFs. Although many target genes have been identified, their functions in cells are hardly analysed and reported in the manuscript. Have authors analysed their functions? Could the analysis provide us with more information on the mechanisms underlying the transition?

1. The rationale of studying FoxO-TFs is a bit vague. Initially, the study was on mTORC1, then GSK3. After showing that neither mTORC1 nor GSK3 accounted for the delayed exit of naïve pluripotency in Pten-KO mESCs, the study started to examine FoxO-TFs without any explanation. Since AKT has many downstream substrates, the authors do not clearly justify why FoxO-TFs became the target to study.

2. AKT phosphorylates FoxO to make them translocate from the nucleus to cytoplasm. In the manuscript, the authors only show the phosphorylation of AKT but do not show that of FoxO. Are there any dynamic changes in FoxO phosphorylation upon withdrawal of 2i in WT mESCs?

3. It is unclear in the manuscript what roles Foxo1/3 play in naïve pluripotency. It shows in Fig 3a that FoxO1 binds to a group of enhancers only in 2i cultures which cannot be detected in Pten-KO mESCs due to a lack of nuclear FoxO1 in these cells (Supplementary Fig 3b). However, it is unclear what function FoxO1 has on these target genes. Have their expression levels been compared in WT vs Pten-KO ESCs under 2i? The comparison may shed light on the function of FoxO1-binding as an activator or repressor and help us to know the role of FoxO-TFs in naïve pluripotency (refer to the discussion). No clear evidence in this manuscript shows that FoxO1 enhances naïve pluripotency.

4. Similarly, what is the dynamic expression of N24-only target genes of FoxO1 (if any) 24 hours after 2i withdrawal? What are their functions?

5. Oct4 and Sox2 have been previously reported to be activated directly by FoxO1 in both mESCs and hESCs (Zhang et al., Nat Cell Biol 2011). In the current ChIP-seq data, are these two genes in the shared group?

Other technical points

1. It might be better that the authors elaborate on why they chose 24 hrs after 2i withdrawal (N24) as the time of interest for most sequencing experiments since there is still a large proportion of Rex1-GFP+ cells (Fig 1b). It would be interesting to see what would happen if both WT and Pten-KO cells are maintained for a longer period.

2. In several figures (e.g. Fig. 1h, 1j, 2f, Suppl Fig. 1i, etc), statistical analysis is performed with only

two independent experiments, which makes significance not meaningful.

3. Fig.2d, it could be clearer if the authors include high magnification images to show dual expression pattern of FoxO1 (nuclear and cytoplasmic), nuclear expression of Otx2 and cytoplasmic expression of Ecad.

Reviewer #2 (Remarks to the Author):

a) Some support for "differentiation-defective ESCs lacking WNT/GSK3 pathway effector Tcf7l1 did not restore Rex1-GFP downstream kinetics upon Rapamycin treatment" (line 112..).

But here the Tcf7l1 KO does not align with the WT control (dashed line in Fig. 1I) and so this experiment is difficult to interpret. It is also not clear how replicates would vary in this plot as only n=1 of the data has been presented.

b)"consistently showing a significantly stronger reduction in expression levels upon Rapamycin treatment in Tsc2 compared to Pten KO (Fig. 1K)" (line 119..). I was not convinced by this analysis of figure 1K. Specifically a large number of fold change values are used in the analysis and this has the potential to generate very low p-values, which are then compared. There is unlikely to be the precision in the experiment to support quantitative comparison of these low p-values. Looking at the plot the overall averages of the fold change values between Tsc2 and Pten KO look almost identical.

Overall I was not convinced by the argument that was being made (line 122 p4)

- that Pten KO phenotype was not exclusively determined by hyperactivity of mTORC1.

c)"Furthermore, 3xFLAG-FoxO1nuc expression in Pten KO ESCs rescues their differentiation defect (Fig 2F..)" (line 167..). Only Nanog and Esrrb are measured here and the controls (especially for nanog) do not seem to completely support the assertion made. This makes interpretation quite difficult.

d)It is confusing how Foxo1 and Foxo3 expression were treated in the analysis. There is an analysis of the overlap of CHIP-Seq peaks, but then the text mentions both FoxO1 and FoxO and FoxO-TF and FoxO TFs as well. This raises a lot of issues, for example is the RNAi specific to FoxO1 or will it also impact the other factors? Can these factors functionally substitute? Understanding the relationship between these gene products would help with interpretation of the data.

e)The FoxO ChIP-seq analysis looks very reasonable for the most part. However, where ChIP-seq overlaps were compared the issue of comparing low p-values comes up again. Specifically, (line 226), "This contrasted with lower overlap between regions bound by FoxO1 in 2i and OCT4 in ESCs (Fig. 3E, F)". This is not really supported by the data as the fold difference is very small between these counts from components of the Venn diagram, p-values are tiny as there is a large count of sites used to make the assertions. But two tiny p-values should

not be compared quantitatively without reference to the size of the effect, especially with ChIP-seq across samples which is a very difficult experiment to control.

f) I did not see much support in the data for "the concomitant presence of FoxO and ESRRB triggers the strongest transcriptional response (Supplementary Fig. 3H, I)" (line 244..).

g)"We found that 2i-specific FoxO1 targets show largely continuous downregulation, N24-specific targets a continuous upregulation..." (line 265..). This effect can be seen in the data (although quite weak). However, since you are effectively sampling genes selected with specific functions (eg 2i-specific) you might well be able to make a similar plot for a random selection of genes, not just FoxO targets. If the overall population of genes has this property then the FoxO targets could be a random sample of genes and still show this property.

Overall I think this is an extremely extensive and detailed study in an interesting scientific area. I think the data is interesting but the data and analysis was not completely convincing as to how FoxO factors fit into the transition from a naïve to a formative state. Some of effects shown were quite modest in magnitude and I think it is important to address this in order to make a convincing argument that this work reveals robust new biology.

Reviewer #3 (Remarks to the Author):

This manuscript studied the role of PI3K/AKT/mTORC1 pathway in the exit of mESC from naïve state. It was previously found that downstream effectors of this pathway such as FOXO transcription factor are associated with the differentiation state of mESC. Although both PTEN KO and TSC2 KO mESC showed defective differentiation, the authors found some differences between the PTEN KO and TSC2 KO cells. Their results suggest that PTEN affect the exit from naïve state by a mechanism independent of mTORC1. They focused on FOXO transcription factors downstream of Akt and found that FOXO1 is translocated to the nucleus during the exit from naïve state due to reduced Akt activity. The authors conducted further studies to show binding of FOXO to genomic regions shared by OCT4, OTX2, and ESRRB. They also found binding of FOXO1 to both naïve marker genes and formative marker genes. Based on these results they concluded that FOXO transcription factors play direct role in the regulation of exit from naïve state. However, these results are mostly correlative.

One major caveat in this study is that in TSC2 KO cells Akt activity is downregulated and therefore FOXO transcription factors are translocated to nucleus. Thus, if FOXO transcription factors play a major role in exit from naïve state, why TSC2 KO cells are impaired in the transition to the differentiated state?

Specific points:

1. The images of FOXO1 in Fig. 2 a, d and supp. Fig. 2 are not very convincing.

2. The authors used transfection to express FLAG-tagged proteins. What is the efficiency of transfection ? Are all the cells transfected? If not, it is difficult to interpret the results.
3. The authors should perform chromatin heatmap after knockdown of FOXO1 and compare gene expression to control cells.
4. Since in TSC2 KO cells FOXO should be nuclear the authors should conduct Chip-seq for FOXO in these cells and compared gene expression level to WT and PTEN KO cells.

We are grateful to the Editor and the three Reviewers for their time and insightful comments on our manuscript, in response to which, we have provided additional data and substantially revised our manuscript.

We now address, in blue text, all of the Reviewers' comments, point-by-point.

REVIEWER COMMENTS

Reviewer #1 (Remarks to the Author):

Naïve and formative pluripotent stem cells that recapitulate the pre- and early post-implantation epiblast, respectively, have distinct epigenetic and gene expression features. Although the core transcriptional networks that maintain these different pluripotent states have been identified, molecular mechanisms control the transition remain unclear. In this manuscript, the authors address this question using *Pten*-KO mESCs that reveal a delayed exit of naïve pluripotency upon differentiation and find that FoxO TFs function to initiate the transition from naïve to formative pluripotency by activating the expression of formative pluripotency-specific genes. They utilized ChIP-seq and RNA-seq techniques to identify the target genes of FoxO1 during the exit of naïve pluripotency. The study is very interesting.

However, the authors seem to mainly focus on the binding sites of FoxO1 rather than the functional impact of the genes bound by FoxO1 except a small number of core GRN TFs. Although many target genes have been identified, their functions in cells are hardly analysed and reported in the manuscript. Have authors analysed their functions? Could the analysis provide us with more information on the mechanisms underlying the transition?

RESPONSE: We thank the Reviewer for the interest in our study and for the suggestion to extend analysis to functions apart from regulation of the formative GRN.

As correctly pointed out, we focused on the intersection of FoxO TFs with the pluripotency network, mainly to explain how these factors could be involved in triggering the initiation of the formative pluripotency specific gene expression programme. This was guided by the differentiation phenotypes of the *Pten* KO and FoxO1 KD cells.

To this end, we performed a number of analyses, including overlaps with known pluripotency-regulating TFs and assessing the impact of perturbing FoxO function. We believe that this strong overlap with the formative pluripotency programme can explain the role of FoxO1 in the exit from naïve pluripotency. However, we agree that studying the role of FoxO-TF targets in more details will provide important additional insights.

To address the Reviewer's comment, we now show KEGG and REACTOME pathway enrichment analysis of genes associated with FoxO1 peaks (*RevFig. 1, Supplementary Fig. 3e*). This analysis showed an enrichment for WNT-signalling and ubiquitin mediated proteolysis in the 2i-specific peaks and for PI3/AKT signalling, focal adhesion and actin cytoskeleton related terms in the N24 specific peaks. Shared peaks were enriched for terms related to general pluripotency and TGF- β signalling.

Several E3-ligases are specifically bound by FoxO1 in the naïve state. These E3-ligases include Mdm2, Fbx17 and Huwe1, and REACTOME pathway analysis of 2i-only bound genes shows an enrichment for the term "Regulation of TP53 Degradation". This could indicate that a role for FoxO1 in ESCs could be maintenance of proper cell cycle control through controlling proteolysis. We now include this information in the discussion section.

RevFig. 1 – KEGG pathway (left) and REACTOME (right) enrichment analysis (EA) on genes associated with indicated FoxO1 peaks (x axis). Top 5 categories enriched in each list are shown on the y axis. Dot colour indicates p-values (only $p \leq 0.1$ are shown). Dot size indicates the GeneRatio (ratio between the overlap size and the category size).

It is very likely that FoxO1 plays roles in addition to triggering the formative GRN, especially the role in naïve pluripotency remains to be worked out in detail. However, 15% of all genes that are differentially expressed between 2i and N24 are FoxO1 targets and more than 40% of FoxO1 targets are DEGs between 2i and N24. Furthermore, new analysis shows that the effect size of FoxO1-dependent gene regulation during differentiation appears on par or larger than that achieved by known pluripotency regulating TFs, such as *Oct4*, *Nanog* and *Otx2* (Fig. 4i). We hope the Reviewer agrees that this is strong motivation to focus on the role of FoxO-TFs in regulating the establishment of formative pluripotent cell identity.

1. The rationale of studying FoxO-TFs is a bit vague. Initially, the study was on mTORC1, then GSK3. After showing that neither mTORC1 nor GSK3 accounted for the delayed exit of naïve pluripotency in *Pten*-KO mESCs, the study started to examine FoxO-TFs without any explanation. Since AKT has many downstream substrates, the authors do not clearly justify why FoxO-TFs became the target to study.

RESPONSE: We apologize for not explaining the rationale to study FoxO TFs more clearly. Indeed, Akt has potentially more downstream targets. However, our decision to look in detail at mTORC1, GSK3 and FoxO-TFs was driven by the fact that all these three are hits in screens for factors that drive the exit from naïve pluripotency (see line 80 of the manuscript).

The focus on FoxO arises from rescue experiments in *Pten* KOs that indicate that mTORC1 deregulation cannot explain the full *Pten* KO phenotype and epistasis experiments that show that in our experimental system GSK3 plays no or only a minor role downstream of Akt. FoxO TFs were the next logical factors to evaluate.

We can of course not exclude other effectors downstream of Akt that contribute to its role in regulating the exit from naïve pluripotency. These are beyond the scope of our study, and we believe that if they exist, their contribution must be relatively small. However, we now acknowledge the possibility that Akt might have regulatory input on functions that we did not assess in our study in the discussion section.

2. AKT phosphorylates FoxO to make them translocate from the nucleus to cytoplasm. In the manuscript, the authors only show the phosphorylation of AKT but do not show that of FoxO. Are there any dynamic changes in FoxO phosphorylation upon withdrawal of 2i in WT mESCs?

RESPONSE: We thank the Reviewer for suggesting this experiment. Investigating the levels of pFoxO1 during differentiation revealed that in parallel to a decrease in pAkt, pFoxO1 levels decrease, while total FoxO1 levels increase during naïve exit. In *Pten* KO cells, pFoxO levels are higher and FoxO levels increase to a lesser extent. This is consistent with other data and the scientific literature and supports a role for Akt in gating nuclear-cytoplasmic localization of FoxO1 by regulating its phosphorylation state. We now include this data shown below as *RevFig. 2* in *Supplementary Fig. 2b* in the manuscript.

RevFig. 2 – Western blot analysis for the indicated proteins in WT and *Pten* KO cells in naïve pluripotency supporting conditions (2i) and 24h (N24) after 2i removal. TUBULIN was used as a loading control.

3. It is unclear in the manuscript what roles Foxo1/3 play in naïve pluripotency. It shows in Fig 3a that FoxO1 binds to a group of enhancers only in 2i cultures which cannot be detected in *Pten*-KO mESCs due to a lack of nuclear FoxO1 in these cells (Supplementary Fig 3b). However, it is unclear what function FoxO1 has on these target genes. Have their expression levels been compared in WT vs *Pten*-KO ESCs under 2i? The comparison may shed light on the function of FoxO1-binding as an activator or repressor and help us to know the role of FoxO-TFs in naïve pluripotency (refer to the discussion). No clear evidence in this manuscript shows that FoxO1 enhances naïve pluripotency.

RESPONSE: We thank the Reviewer for bringing up this point and for the suggested experiment. We do not wish to make the claim that FoxO1 enhances naïve pluripotency. However, our data indicate that FoxO1 might be required for maintaining naïve pluripotency. For a discussion of a potential function of genes bound by Foxo1 in 2i, please see *RevFig 1* and related discussion above.

We have further performed the analysis suggested by the Reviewer to investigate whether FoxO1 2i targets are specifically deregulated in *Pten* KO in 2i. Overall, FoxO1 2i target genes showed a significantly stronger downregulation than non-targets in *Pten* KO ESCs (now shown in *Supplementary Fig. 4j, k; RevFig. 3*). This is consistent with a role for FoxO1 to drive expression of its target genes. However, a large portion of FoxO1 targets also showed upregulation in *Pten* KO, where nuclear FoxO levels are strongly reduced. This is consistent with a dual role for FoxO1 in activating and repressing transcription. This dual role is consistent with our data, but also with multiple other lines of evidence shown in the manuscript and the literature, which we summarize in the discussion section.

*RevFig. 3 – (a) Distribution of log₂FC of differentially expressed genes (DEGs, $p \leq 0.05$) in *Pten* KO cells as compared to WT cells in 2i, divided in FoxO1 2i targets (green) and non-targets (grey) (b) Box plot showing the expression of DEGs ($p \leq 0.05$) in *Pten* KO cells in 2i, divided into FoxO1 2i targets (green) or non-targets (grey), as measured by RNA-seq. Data is shown as log₂FC relative to WT. The resulting p-value from two-tailed Wilcoxon rank sum test is indicated in the plot.*

4. Similarly, what is the dynamic expression of N24-only target genes of FoxO1 (if any) 24 hours after 2i withdrawal? What are their functions?

RESPONSE: In *Figs 4a, b* and *Supplementary Fig. 4a* and *4c* we address the point raised by the Reviewer. N24-only targets are largely genes that show transcriptional upregulation during naïve exit. They are enriched for terms related to transcriptional regulation, DNA binding, protein phosphorylation and pluripotency. This suggests that FoxO1 TFs regulate genes that are themselves transcriptional regulators, placing FoxO1 on top of a functional cascade to trigger a formative pluripotent gene expression state.

5. Oct4 and Sox2 have been previously reported to be activated directly by FoxO1 in both mESCs and hESCs (Zhang et al., Nat Cell Biol 2011). In the current ChIP-seq data, are these two genes in the shared group?

RESPONSE: In our analysis, Oct4 is a N24-specific FoxO target, whereas Sox2 is not a target. However, closer inspection of ChIP-seq tracks shows an accumulation of FoxO1 on a H3K27ac decorated Sox2 enhancer, specifically at N24 (*RevFig. 4*). Any cutoff based analysis will suffer from false negatives. We deliberately chose a cutoff that generates a high confidence set of FoxO1 target genes, hence we might have missed the relatively low but clear enrichment on the Sox2 locus. We note that Zhang et al. have performed ChIP qPCR analysis on a small set of target promoters. These assays are able to detect small enrichments on individual loci. Our analysis is more global and relatively small enrichments on individual

loci will not be included in lists of highly significant ChIP-seq targets, based on applied cut-offs.

RevFig. 4 – IGV browser snapshot showing the chromatin state of the Sox2 (left) and Oct4 (right) locus. The location of the FoxO1 peak is highlighted with a box.

Other technical points

1. It might be better that the authors elaborate on why they chose 24 hrs after 2i withdrawal (N24) as the time of interest for most sequencing experiments since there is still a large proportion of Rex1-GFP+ cells (Fig 1b). It would be interesting to see what would happen if both WT and Pten-KO cells are maintained for a longer period.

RESPONSE: Our reasoning for choosing the 24h time point is that at this time point cells are in the process of making irreversible differentiation decisions. 48h after the onset of differentiation all WT cells have shut down Rex1-GFP expression and the large majority of cells have irreversibly exited the naïve state. The completion of Rex1-GFP downregulation at N48 suggests that the mechanisms driving naïve exit might no longer be in operation at N48, but they should be at maximum activity at N24. Hence, we believe that 24h after the onset of differentiation captures the time point of cellular decision-making best. We now more carefully explain this reasoning in the results section.

Of course, the role of Pten and FoxO1 beyond the first 24h of differentiation, possibly during definitive lineage choice is a very interesting question, but one for future studies.

2. In several figures (e.g. Fig. 1h, 1j, 2f, Suppl Fig. 1i, etc), statistical analysis is performed with only two independent experiments, which makes significance not meaningful.

RESPONSE: To strengthen our claims, we have performed further replicates for Figs. 1h (now Supplementary Fig. 1i), Supplementary Fig. 1i, 2f (now Supplementary Fig. 2i), Supplementary Fig. 2h (now Supplementary Fig. 2i).

Fig. 1j shows DEseq-normalized counts based on RNA-seq experiments performed in duplicates. We believe that this representation provides a more quantitative information compared to heatmaps that could also be used to represent RNA-seq results.

3. Fig.2d, it could be clearer if the authors include high magnification images to show dual expression pattern of FoxO1 (nuclear and cytoplasmic), nuclear expression of Otx2 and cytoplasmic expression of Ecad.

RESPONSE: We now show higher quality images in Fig 2d. Furthermore, we have expanded this analysis and have generated important additional data.

We now show that a transient nuclear localization of FoxO1 at the time when formative cell identity is acquired is also a feature of *in vivo* epiblast maturation. Specifically, we showed that in rosette-stage embryos at E4.75, FoxO1 localizes to the nucleus significantly more than in E4.5 and E5.5 epiblast (RevFig. 5). Furthermore, genes that are differentially expressed in the naïve to formative and primed transition *in vivo* are significantly enriched for FoxO1 targets. We now show these data in Fig. 2d, e, Supplementary Fig. 2c and Fig. 4h. Together, this significantly strengthens the manuscript and indicates that FoxO1 nuclear translocation is also a feature of the *in vivo* pre- to post-implantation epiblast transition.

RevFig. 5 Confocal microscopy images after IF showing FOXO1 (white), SOX2 (green), GATA4 (red), and OTX2 (red) and ECAD (green) in WT E4.5, E4.75 and E5.5 embryos. Hoechst staining is shown in cyan. One representative image of n=3 independent experiments is shown. Scale bar = 100 μ M.

Reviewer #2 (Remarks to the Author):

a) Some support for "differentiation-defective ESCs lacking WNT/GSK3 pathway effector Tcf7l1 did not restore Rex1-GFP downstream kinetics upon Rapamycin treatment" (line 112..).

But here the Tcf7l1 KO does not align with the WT control (dashed line in Fig. 1I) and so this experiment is difficult to interpret. It is also not clear how replicates would vary in this plot as only n=1 of the data has been presented.

RESPONSE: We apologize for not describing this experiment appropriately. The dashed line shows WT cells treated with Rapamycin after 24h of differentiation. This sample is shown in all panels as reference.

Tcf7l1 KOs serve as a control that is resistant to Rapamycin-induced rescue of differentiation, showing the specificity of the experimental setup. Hence, the Tcf7l1 KO shows high Rex1-GFP levels in both Rapamycin treated and untreated samples. Overall, this experiment supports a specific role of mTORC1 deregulation downstream of Pten and Tsc2 KOs. It further shows that Rapamycin induced rescue is weaker in Pten compared to Tsc2 KOs. As described in the Figure legend, we had performed several additional replicates of this experiment with overlapping results, which are now shown below (RevFig. 6).

RevFig. 6 – Additional replicates for experiments shown in Fig. 1h.

b) "consistently showing a significantly stronger reduction in expression levels upon Rapamycin treatment in Tsc2 compared to Pten KO (Fig. 1K)" (line 119.) . I was not convinced by this analysis of figure 1K. Specifically a large number of fold change values are used in the analysis and this has the potential to generate very low p-values, which are then compared. There is unlikely to be the precision in the experiment to support quantitative comparison of these low p-values. Looking at the plot the overall averages of the fold change values between Tsc2 and Pten KO look almost identical. Overall I was not convinced by the argument that was being made (line 122 p4)

- that Pten KO phenotype was not exclusively determined by hyperactivity of mTORC1.

RESPONSE: We agree that a direct comparison of low *p-values* would not allow precise comparative statements. We do not want to make such claims based on comparing significance levels and our conclusions are not based on such analyses.

The data in previous *Fig. 1k* (now *Fig. 1j*) shows that whereas the upregulation of naïve genes is not different between *Pten* and *Tsc2* KO ESCs at N24, a significant difference is established ($p=0.0035$) at the population level of ~400 naïve pluripotency specific genes upon Rapamycin treatment. This means that *Pten* KOs still express significantly higher levels of naïve genes compared to *Tsc2* mutants after Rapamycin treatment, supporting the conclusion that mechanisms other than mTORC1 contribute to naïve gene deregulation in *Pten* KOs.

We have further analyzed the amplitude of expression changes during the naïve to formative transition in *Pten* and *Tsc2* mutants and compared expression changes in the presence of DMSO and Rapamycin. This analysis showed that *Tsc2* mutants downregulate naïve genes statistically significantly more during the naïve to formative transition after Rapamycin treatment compared to *Pten* KOs (*Fig. 1k*).

In addition, we show two additional replicates of FACS-based experiments as *RevFig. 7* that show that whereas the differentiation delay of *Tsc2* KO ESCs is completely rescued by Rapamycin, that of *Pten* KOs persists even in presence of Rapamycin and the rescue is only partial and clearly less strong than in *Tsc2* KOs.

We hope that this multilayered evidence based on multiple FACS experiments and RNA-seq convinces the Reviewer, that the *Pten* KO phenotype of naïve exit defects cannot be completely rescued by Rapamycin treatment and is therefore not exclusively mediated by mTORC1 deregulation.

*RevFig 7 – (a) Additional FACS experiments showing Rapamycin mediated rescue of *Pten* and *Tsc2* mutant ESCs (b) Box plot showing the expression of naïve genes (combination of naïve early and naïve late genes as defined in Carbognin et al. 2023) in *Pten* and *Tsc2* KO cells treated with Rapamycin as measured by RNA-seq. Data is shown as log₂FC relative to DMSO control. The resulting *p*-value from two-tailed Wilcoxon signed rank test is indicated in the plot.*

c) "Furthermore, 3xFLAG-FoxO1nuc expression in *Pten* KO ESCs rescues their differentiation defect (Fig 2F...)" (line 167..). Only *Nanog* and *Esrrb* are measured here and the controls (especially for *nanog*) do not seem to completely support the assertion made. This makes interpretation quite difficult.

RESPONSE: Thank you for pointing out this issue. We have now performed qPCRs on additional samples from independent experiments. Results are shown as *Supplementary Fig. 2l* in the revised manuscript (*RevFig. 8*). Together, the FACS and qPCR data shown in *Supplementary Fig. 2k and 2l* show that the *Pten* KO phenotype can be rescued, at least in part, by enforcing nuclear FoxO1 localization in a dose dependent manner. Interestingly, overexpression of nuclear FoxO1 in *Pten* KO cells had distinct results on different naïve markers: While it reduced the expression of *Nanog* and *Esrrb*, in line with the global rescue of the differentiation defect of *Pten* mutants, it upregulated the expression of *Klf5*. This is consistent with a target-specific activator or silencer function of FoxO1.

RevFig. 8 Expression levels of *Foxo1*, *Nanog*, *Esrrb*, and *Klf5* measured by RT-qPCR in *Pten* KO cells expressing 3xFLAG-FoxO1nuc (*P-F^{nuc}*) at N24 after 8 hours treatment with 500 ng/ml doxycycline (orange). Control WT cells and *Pten* KO cells are also included (grey). Mean and SD of *n*=3 independent experiments (distinguished by distinct shades of grey) are shown. Expression was normalised to *Rpl32*, and corresponding -dox samples for each cell line were set to one (indicated by dashed line). *p*-values show results of paired, two-tailed *t*-tests, between -dox and +dox.

d) It is confusing how Foxo1 and Foxo3 expression were treated in the analysis. There is an analysis of the overlap of ChIP-Seq peaks, but then the text mentions both FoxO1 and FoxO and FoxO-TF and FoxO TFs as well. This raises a lot of issues, for example is the RNAi specific to FoxO1 or will it also impact the other factors? Can these factors functionally substitute? Understanding the relationship between these gene products would help with interpretation of the data.

RESPONSE: Our reasoning for referring to FoxO-TFs without further specification was that FoxO1 and FoxO3 show a largely overlapping ChIP-seq profiles. However, we agree with the Reviewer that the general terminology used in most downstream analyses that have been performed with FoxO1 is suboptimal. To avoid confusion, we therefore revised the manuscript and now refer to specific FoxO factors when describing results.

The Reviewer raises the interesting point of a potential redundancy between FoxO TFs. Analysing redundancies between FoxO factors is complicated by the difficulty to establish *FoxO1* KO cell lines. siRNA experiments give some hints towards the complexity of redundancy between FoxO factors. Depletion of FoxO3 and FoxO4 by siRNA results in increased FoxO1 expression, whereas siFoxO3 does not increase FoxO1 levels. Moreover, when using double siRNA against FoxO1 and FoxO3 we cannot achieve complete depletion of FoxO1 (which is possible in single siRNA experiments). Together this indicates some crosstalk between FoxO1 proteins and indicates strong selective pressure to maintain some

FoxO1 expression (*RevFig. 9*). We hope that the Reviewer agrees that the cooperativity and potential redundancy between FoxO-family TFs are beyond the scope of this study.

RevFig. 9 – Western blot analysis for FoxO1 in WT cells treated with control (siScr, siGFP) or siRNA against FoxO1 (siFoxO1), FoxO3 (siFoxO3) or FoxO4 (siFoxO4) 24h (N24) after 2i removal. VINCULIN was used as a loading control.

e) The FoxO ChIP-seq analysis looks very reasonable for the most part. However, where ChIP-seq overlaps were compared the issue of comparing low p-values comes up again. Specifically, (line 226), "This contrasted with lower overlap between regions bound by FoxO1 in 2i and OCT4 in ESCs (Fig. 3E, F)". This is not really supported by the data as the fold difference is very small between these counts from components of the Venn diagram, p-values are tiny as there is a large count of sites used to make the assertions. But two tiny p-values should not be compared quantitatively without reference to the size of the effect, especially with ChIP-seq across samples which is a very difficult experiment to control.

RESPONSE: We agree that a difference between already small *p-values* should not be interpreted as a significant difference between comparisons without reference to the effect size. We now clarify that our statement is based on the fact that the overlap between 2i-bound FoxO1 peaks and ESC-specific OCT4 bound enhancers is 4%, whereas this percentage increases to 12% for FoxO1 peaks at N24 and EpiLC specific Oct4 bound enhancers. We have updated the manuscript text accordingly. To better describe experiments shown in Figure 3, we have carefully rewritten the corresponding results section and hope to have increased clarity and precision.

f) I did not see much support in the data for "the concomitant presence of FoxO and ESRRB triggers the strongest transcriptional response (Supplementary Fig. 3H, I)" (line 244..).

RESPONSE: We apologize for this unintentional misinterpretation. The heatmaps in *Supplementary Fig. 3i* indeed only show enhancer marks and not transcriptional activity. We change the text to: "Loci exhibiting a concomitant presence of FoxO1 and ESRRB show stronger enhancer activity compared to loci bound by only FOXO1 or ESRRB." This statement is supported by the overall higher levels of enhancer marks on shared targets.

g) "We found that 2i-specific FoxO1 targets show largely continuous downregulation, N24-specific targets a continuous upregulation..." (line 265..). This effect can be seen in the data

(although quite weak). However, since you are effectively sampling genes selected with specific functions (eg 2i-specific) you might well be able to make a similar plot for a random selection of genes, not just FoxO targets. If the overall population of genes has this property then the FoxO targets could be a random sample of genes and still show this property.

RESPONSE: We thank the Reviewer for pointing this out and for the suggested experiment. In our comparison, all non-targets served as control-set, but to avoid potential biases based on random sampling, we performed the analysis as suggested by the Reviewer. To this end we have performed 30,000 random draws of gene-sets of the same size as the sample gene sets and have plotted the distribution of log2 fold changes (log2FC). Based on these distributions, we then calculated the empirical *p-value* for the observed log2FC in FoxO1 target gene groups. As can be seen in *RevFig. 10* (new *Fig. 4b* in the manuscript), 2i-specific, N24-specific and shared FoxO1 targets are all highly significant outliers compared to randomly selected groups of genes of the same size. This shows that FoxO1 binding is strongly correlated to and a potential regulator of gene expression changes during differentiation.

We hope that this analysis can convince the Reviewer that the observed effects are significant and by no means weak.

RevFig. 10 – Bootstrapping analysis of 30,000 draws of gene-sets of the same size as the sample gene sets (2i-specific, 249 genes; N24-specific, 889 genes; 2i&N24, 486 genes). The resulting distribution of average log2FC is plotted. Based on these distributions, we then calculated the empirical p-value for the observed log2FC changes in FoxO1 target gene groups. p-values that indicate the probability of the predicted effect- size exceeding the measured effect size are added next to the corresponding plots.

We have further changed the visualization of gene expression dynamics of 2i-specific, 2i&N24 and N24-specific FoxO target genes across a 2h-resolved 32h long differentiation time course. These now show better that early in differentiation, N24-specific FoxO1 targets are up- and 2i-specific targets downregulated. These data are now shown in *Supplementary Fig. 4c*.

Overall I think this is an extremely extensive and detailed study in an interesting scientific area. I think the data is interesting but the data and analysis was not completely convincing as to how FoxO factors fit into the transition from a naive to a formative state.

Some of effects shown were quite modest in magnitude and I think it is important to address this in order to make a convincing argument that this work reveals robust new biology.

RESPONSE: We thank the Reviewer for appreciating the detailed nature and the general interest of our study. The Reviewer raises the important point about the magnitude of the observed effects. Based on the relatively low overall expression changes within large target gene groups, we can understand this comment. However, two points need to be considered when considering the biological significance of our findings: Firstly, a small impact on individual genes will add up if, such as in case of FoxO-TFs, regulation is performed on hundreds of target genes that belong to a common cellular function (in this case the initiation of the formative GRN). Secondly, effect size needs to be seen in relative terms. How strong is the effect of FoxO1 in relation to other known TFs that regulate naïve and formative pluripotency? To address the question of effect size, we have performed several experiments shown in in *Fig 4b, l* and in *Supplementary Fig. f, m (RevFig. 11)*. These experiments were inspired by this Reviewer's comment (g).

Strikingly, we can show through bootstrapping experiments that FoxO1 targets are more strongly transcriptionally regulated during naïve exit than targets of key pluripotency TFs such as Oct4, Sox2, Esrrb or Tcf7l1. Only Otx2 or Tcf7l1 binding is a similarly good predictor to gene expression changes. (*RevFig. 11 a*).

However, even within Otx2 and Tcf7l1 targets, FoxO1/Otx2 or FoxO1/Tcf7l1 shared target genes have a significantly higher amplitude of gene regulation during differentiation (*RevFig. 11 b, c*). We show these new key data in *Fig. 4b, i* and *Supplementary Fig. 4l, m*.

These results show that the FoxO1 binding is, alone or in combination with known pluripotency regulators, a better predictor for gene expression changes during differentiation than individual known pluripotency TFs. We also argue that these analyses show that the effect size is in fact not small or modest. Together, this positions FoxO1 as a key pluripotency TF.

RevFig 11 – (a) Bootstrapping analysis of 30,000 draws of gene-sets of the same size as the sample gene sets (pluripotency TF targets, n indicates size of gene group). The resulting distribution of average \log_2FC is plotted. Based on these distributions, we then calculated the empirical p -value for the observed \log_2FC in indicated FoxO1 target gene groups. p -values that indicate the probability of the predicted effect size exceeding the measured effect size are stated within the corresponding plots **(b)** Box plot showing the expression of genes upregulated during WT differentiation (at N24), that are either co-bound by FoxO1 and Otx2 (pink), bound only by FoxO1 (purple), bound only by Otx2 (dark grey), or not bound (light grey). Data is shown as \log_2FC relative to WT. FoxO1 N24 peaks and Otx2 EpiLC peaks (as described in Fig.3) were considered. The resulting p -values from two-tailed Wilcoxon rank sum test comparing each group to the co-bound group are indicated in the plot **(c)** Box plot showing the expression of genes downregulated during WT differentiation (at N24), that are either shared targets of FoxO1 and Tcf711 in 2i (pink), only FoxO1 2i targets (purple), only Tcf711 2i targets (dark grey), or non-targets (light grey). Data is shown as \log_2FC relative to WT. The resulting p -values from two-tailed Wilcoxon rank sum test comparing each group to shared target group are indicated in the plot.

Reviewer #3 (Remarks to the Author):

This manuscript studied the role of PI3K/AKT/mTORC1 pathway in the exit of mESC from naïve state. It was previously found that downstream effectors of this pathway such as FOXO transcription factor are associated with the differentiation state of mESC. Although both PTEN KO and TSC2 KO mESC showed defective differentiation, the authors found some differences between the PTEN KO and TSC2 KO cells. Their results suggest that PTEN affect the exit from naïve state by a mechanism independent of mTORC1. They focused on FOXO transcription factors downstream of Akt and found that FOXO1 is translocated to the nucleus during the exit from naïve state due to reduced Akt activity. The authors conducted further studies to show binding of FOXO to genomic regions shared by OCT4, OTX2, and ESRRB. They also found binding of FOXO1 to both naïve marker genes and formative marker genes. Based on these results they concluded that FOXO transcription factors play direct role in the regulation of exit from naïve state. However, these results are mostly correlative.

RESPONSE: Indeed, genome-wide binding analyses, and analysis of the impact of FoxO binding on global gene expression are to a large extent correlative by nature. However, our manuscript contains multiple functional experiments, where we perturb FoxO1 levels by siRNA or FoxO1 localization in *Pten* KOs by MK-2206 treatment. All of these experiments are consistent with a crucial and previously unrecognized role of FoxO TFs in regulating pluripotency transitions. Importantly, we have now extended our *in vivo* analyses and found that, similar to the exit from naïve pluripotency in culture, FOXO1 transiently translocates to the nucleus in rosette-stage embryos at E4.75 (see *RevFig. 13*). Consistent with a role for FoxO1 in facilitating the naïve to formative transition *in vivo*, FoxO1 target genes show a significantly stronger amplitude of gene regulation compared to non-target genes (see *Fig. 2d, e Supplementary Fig. 2c, Fig. 4h* and response to specific point 1 below).

We also want to point out the effect size analysis performed in response to Reviewer 2 (*RevFig. 11* and *Fig. 4b, i* and *Supplementary Fig. 4l, m*). These analyses show that FoxO1 binding predicts larger changes in gene regulation than that of known pluripotency regulators such as Oct4, Nanog, Esrrb and β -catenin and on par with factors such as Otx2. In sum, our work provides strong evidence for a previously uncharacterized role of FoxO TFs in regulating the key cell fate decision to exit the naïve pluripotent state.

One major caveat in this study is that in TSC2 KO cells Akt activity is downregulated and therefore FOXO transcription factors are translocated to nucleus. Thus, if FOXO transcription factors play a major role in exit from naïve state, why TSC2 KO cells are impaired in the transition to the differentiated state?

RESPONSE: In parallel to the analysis shown in *Fig. 2c*, we had analyzed FoxO1 localisation in *Tsc2* KO cells. Results are now shown in *RevFig. 12*. In our analysis, we cannot detect elevated nuclear FoxO1 localisation in *Tsc2* KOs, neither in ESCs nor in early post-implantation like epiblast-like cells at N24 (*RevFig. 12a, b*).

We acknowledge the Reviewer's comments regarding the potential feedback loop from mTORC1 to Akt. This could occur e.g. via regulation of IRS1 stability, as suggested by the literature (doi.org/10.1016/j.isci.2018.06.006). According to this mechanism, increased mTORC1 activity could result in reduced insulin signaling-driven pAkt levels via negative regulation of IRS1 stability. However, whether this feedback loop is active during the exit from naïve pluripotency remains uncertain. Additionally, the significance of insulin signaling as a

primary functional contributor to pAkt levels during *in vitro* differentiation and peri-implantation development is also unclear. In our *in vitro* system, for example, insulin levels are identical between naïve media and differentiation media.

Pathway crosstalk and feedback circuits downstream of Akt are intriguing questions but, while absolutely important, are not the main focus of our current study and do not directly impact the principal conclusions we present. Nevertheless, our data clearly demonstrate that in *Tsc2* mutant ESCs within our *in vitro* model system, there is no increased localization of FoxO1 to the nucleus during the exit from naïve pluripotency. In fact, we observe a reduction in nuclear FOXO1 upon *Tsc2* depletion. Furthermore, our Western blot analysis indicates that pAkt levels do not undergo substantial changes following *Tsc2* depletion (*RevFig. 12c*).

We hope the Reviewer agrees that our findings effectively support our conclusions and that FOXO1 localization and pAkt levels are not crucially impacted by loss of *Tsc2* in the context of our study.

RevFig 12 – (a) IF analysis using indicated antibodies in 2i and at N24. (b) Quantification of nuclear FoxO1 signal in indicated cell lines in 2i and at N24. (c) Western blot analysis in WT, Pten KO and Tsc2 KO cells in naïve pluripotency supporting conditions (2i). VINCULIN was used as a loading control.

Specific points:

1. The images of FOXO1 in Fig. 2 a, d and supp. Fig. 2 are not very convincing.

RESPONSE: We now show higher quality images in *Fig. 2d*. Furthermore, we have expanded this analysis and have generated important additional data.

We now show that a transient nuclear localization of FoxO1 at the time when formative cell identity is acquired is also a feature of *in vivo* epiblast maturation. Specifically, we showed that in rosette-stage embryos at E4.75, FoxO1 localizes to the nucleus significantly more than in E4.5 and E5.5 epiblast. Furthermore, genes that are differentially expressed in the naïve to formative and primed transition *in vivo* are significantly enriched for FoxO targets. We now show these data in *Fig. 2d, e, Supplementary Fig. 2c* and *Fig. 4h*. Together, this significantly strengthens the manuscript and indicates that FoxO1 nuclear translocation is a feature of the *in vivo* pre- to post-implantation epiblast transition.

RevFig. 13 – Confocal microscopy images after IF showing FOXO1 (white), SOX2 (green), GATA4 (red), and OTX2 (red) and ECAD (green) in WT E4.5, E4.75 and E5.5 embryos. Hoechst staining is shown in cyan. One representative image of n=3 independent experiments is shown. Scale bar = 100 μM.

We are not certain what is the issue with *Fig. 2a* and *Supplementary Fig. 2d*. They are of high image quality. The images shown in *Fig. 2a* and others were used to quantify FOXO1 and ESRRB levels. We believe that the data shown is sufficient to fully support the conclusions drawn from this experiment and hope that the Reviewer agrees with us.

2. The authors used transfection to express FLAG-tagged proteins. What is the efficiency of transfection? Are all the cells transfected? If not, it is difficult to interpret the results.

RESPONSE: Our experiments were performed in stable clonal lines expressing the FoxO1 transgenes in a dox-inducible manner. The images in *Supplementary Fig. 2i* show that whereas without dox, all cells express rather homogeneous Nanog, cells expressing FoxO1 express substantially lower Nanog levels. This supports the conclusion that FoxO1 overexpression leads to extinction of naïve pluripotency.

3. The authors should perform chromatin heatmap after knockdown of FOXO1 and compare gene expression to control cells.

RESPONSE: We thank the Reviewer for this suggestion. It is indeed important to show that a reduction of FoxO-TFs leads to a reduction of H3K27ac on formative enhancers. We have already shown that the upregulation of FoxO1 target genes is not occurring appropriately in FoxO1 knockdown cells (*Fig. 5b*).

To evaluate whether loss of FoxO1 indeed impacts enhancer activation of formative genes, we have now performed CUT&RUN analysis after treatment with Foxo1 siRNA. Results show that all tested enhancers show a clear low H3K27ac signal upon siRNA-mediated depletion of FoxO1. These results are shown in *Fig. 5c* and *Supplementary Fig. 5g*. We conclude from these experiments that FoxO1 is required for proper enhancer activation of formative genes at the exit from naïve pluripotency.

RevFig 15 – CUT&RUN analysis of H3K27ac levels on indicated enhancers after FoxO1 siRNA. Signal was normalized to a genomic background region that did not exhibit any H3K27ac signal in WT cells. Data was further normalized to a H3K27ac peak found in Drosophila spike-in cells. Data is shown as relative to siScr control. n=2 biological replicates.

4. Since in TSC2 KO cells FOXO should be nuclear the authors should conduct Chip-seq for FOXO in these cells and compared gene expression level to WT and PTEN KO cells.

RESPONSE: Here we refer to our results shown above in *RevFig. 13*. As FoxO1 does not appear to show a stronger nuclear localization in *Tsc2* KOs, we believe that this experiment is not required to support our conclusions. We hope the Reviewer agrees to this point.

REVIEWERS' COMMENTS

Reviewer #1 (Remarks to the Author):

The authors have addressed all my questions and I am satisfied with most of their changes in the revised manuscript. However, there is still one confusion on the new supplementary Fig 3e. The authors stated in the text that 'KEGG and REACTOME pathway enrichment analyses (EA) showed an enrichment for WNT-signalling and ubiquitin mediated proteolysis in the 2i-specific peaks' (Line 197). However, in the figure, the peaks appeared in both 2i-specific and shared column. Please clarify this.

Reviewer #3 (Remarks to the Author):

The authors have addressed my previous comments.

Reviewer #1 (Remarks to the Author):

The authors have addressed all my questions and I am satisfied with most of their changes in the revised manuscript. However, there is still one confusion on the new supplementary Fig 3e. The authors stated in the text that 'KEGG and REACTOME pathway enrichment analyses (EA) showed an enrichment for WNT-signalling and ubiquitin mediated proteolysis in the 2i-specific peaks' (Line 197). However, in the figure, the peaks appeared in both 2i-specific and shared column. Please clarify this.

R: We are happy that the Reviewer appreciates our efforts to improve the manuscript.

To address the above point and to remove ambiguity regarding an enrichment for Wnt-signalling (which is indeed enriched in 2i-only and shared peaks), we have amended the text to: "KEGG and REACTOME pathway enrichment analyses (EA) showed an enrichment for ubiquitin mediated proteolysis in the 2i-specific peaks and for PI3/AKT signalling, focal adhesion and actin cytoskeleton related terms in the N24 specific peaks."

Reviewer #3 (Remarks to the Author):

The authors have addressed my previous comments.

R: Thank you for your suggestions and for appreciating our efforts to improve the manuscript based on your suggestions.